# A stalled-ribosome rescue factor Pth3 is required for mitochondrial translation against antibiotics in *Saccharomyces cerevisiae*

Soichiro Hoshino[1], Ryohei Kanemura[1], Daisuke Kurita[2], Yukihiro Soutome[1], Hyouta Himeno[2], Masak Takaine [3,4], Masakatsu Watanabe [5] & Nobukazu Nameki [1✉]

Mitochondrial translation appears to involve two stalled-ribosome rescue factors (srRFs). One srRF is an ICT1 protein from humans that rescues a "non-stop" type of mitochondrial ribosomes (mitoribosomes) stalled on mRNA lacking a stop codon, while the other, C12orf65, reportedly has functions that overlap with those of ICT1; however, its primary role remains unclear. We herein demonstrated that the *Saccharomyces cerevisiae* homolog of C12orf65, Pth3 (Rso55), preferentially rescued antibiotic-dependent stalled mitoribosomes, which appear to represent a "no-go" type of ribosomes stalled on intact mRNA. On media containing a non-fermentable carbon source, which requires mitochondrial gene expression, respiratory growth was impaired significantly more by the deletion of *PTH3* than that of the *ICT1* homolog *PTH4* in the presence of antibiotics that inhibit mitochondrial translation, such as tetracyclines and macrolides. Additionally, the in organello labeling of mitochondrial translation products and quantification of mRNA levels by quantitative RT-PCR suggested that in the presence of tetracycline, the deletion of *PTH3*, but not *PTH4*, reduced the protein expression of all eight mtDNA-encoded genes at the post-transcriptional or translational level. These results indicate that Pth3 can function as a mitochondrial srRF specific for ribosomes stalled by antibiotics and plays a role in antibiotic resistance in fungi.

[1] Division of Molecular Science, Graduate School of Science and Technology, Gunma University, Kiryu, Gunma, Japan. [2] Department of Biochemistry and Molecular Biology, Faculty of Agriculture and Life Science, Hirosaki University, Hirosaki, Japan. [3] Gunma University Initiative for Advanced Research (GIAR), Gunma University, Maebashi, Japan. [4] Institute for Molecular and Cellular Regulation, Gunma University, Maebashi, Gunma, Japan. [5] Graduate School of Frontier Biosciences, Osaka University, Suita, Osaka, Japan. ✉email: nameki@gunma-u.ac.jp

During translation elongation, ribosome stalling often occurs unexpectedly for a number of endogenous and exogenous reasons. Therefore, the translation system in not only cells, but also organelles appears to be equipped with more than one stalled-ribosome rescue factor (srRF). One of the endogenous reasons for unexpected ribosome stalling is that ribosomes translate an incomplete mRNA lacking a stop codon, namely, a non-stop mRNA, which is generated due to premature transcription termination by RNA polymerase, or RNA digestion by RNase[1]. Consequently, ribosomes are stalled at the 3′ end of the truncated mRNA, leaving both the A-site and mRNA entry channel of ribosomes vacant. This type of stalled ribosome is termed a "non-stop" ribosome. In *Escherichia coli*, various in vivo and in vitro studies have shown that non-stop ribosomes are efficiently rescued by three srRFs, namely, a complex of tmRNA and the SmpB protein, ArfA protein, and YaeJ (ArfB) protein[2–4]. Although the sequences and structures of the three srRF proteins share no homology, recent structural analyses revealed a common mechanism by which these proteins bind to the vacant A-site of non-stop ribosomes[5]. The mRNA entry channel downstream of the A-site is recognized by the C-terminal region of each of these proteins, as described later.

Mitochondria have their own translation system that has a similar architecture to that of its bacterial relatives, but exhibits a number of organelle-specific features. Mitochondrial ribosomes (mitoribosomes) synthesize a limited number of proteins, the genes of which are encoded by mitochondrial DNA (mtDNA) and differ in number among species (for example, eight protein genes in yeast and 13 in humans). All or most of the protein products are membrane proteins that function as core subunits of enzyme complexes of the oxidative phosphorylation (OXPHOS) system[6]. In protein synthesis, nascent polypeptides are inserted into the inner membrane in a co-translational process, and associate with nuclear-encoded subunits according to prearranged procedures that differ among individual complexes[6–8]. Previous studies demonstrated that two proteins play a key role in the rescue of stalled mitoribosomes in mitochondrial translation (Table 1). One is the eukaryotic YaeJ homolog, the ICT1 protein, which was initially identified as srRF in mitochondria from human cells[9]. ICT1 belongs to the RF-1 family, as represented by class I peptide release factors (RFs) from bacteria and mitochondria[10]. This family is characterized by a catalytic structural domain containing a conserved Gly–Gly–Gln (GGQ) motif, termed the GGQ domain (Supplementary Fig. 1). This domain interacts with the peptidyltransferase center of the large ribosomal subunit to trigger peptidyl-tRNA hydrolysis (PTH), resulting in the release of the nascent polypeptide chain from P-site-bound peptidyl-tRNA. In vitro translation experiments using ribosomes

from *E. coli* or pig liver mitochondria showed that ICT1 hydrolyzes peptidyl-tRNA via the GGQ motif on ribosomes stalled by non-stop mRNAs as efficiently as YaeJ[11–14]. The knockdown of ICT1 in HeLa cells results in a reduction in the protein expression of all mtDNA genes, leading to a severe loss of cell viability as well as mitochondrial dysfunction[9,10]. A characteristic feature of the ICT1/YaeJ homolog subfamily is the C-terminal extension following the GGQ domain, which is unstructured in its free form and has an abundance of basic residues[11]. The crystal structure of YaeJ bound to the *Thermus thermophilus* 70S ribosome in complex with initiator tRNA$^{fMet}_i$ and a short non-stop mRNA provided insights into the function of the C-terminal extension on non-stop ribosomes[15]. When YaeJ is bound to the ribosome, a section of its C-terminal extension (termed the C-terminal tail) forms a helical structure, which lies in the mRNA entry channel, downstream of the A-site that is vacant in the 30S subunit. Accordingly, the C-terminal tail is thought to act as a sensor that discriminates between stalled and actively translating ribosomes. Residues involved in the interactions of the C-terminal tail with the ribosome are highly conserved among proteins belonging to the ICT1/YaeJ homolog subfamily of various species[11], and YaeJ and ICT1 were previously shown to be functionally interchangeable between *Caulobacter crescentus* and HEK293 cells[14]. Hence, the C-terminal tail of ICT1 as well as the GGQ domain is most likely to act on stalled mitoribosomes, as observed in the co-crystal structure, despite substantial differences in the components of ribosomes, except for the catalytic regions. Additionally, previous studies showed that ICT1 tightly binds to the large subunit of mitoribosomes as a ribosomal component;[9,16] however, the bound form of ICT1 did not appear to contribute to the rescue of stalled ribosomes[13].

Although the C12orf65 protein has been proposed as a second srRF, its primary role remains unclear. C12orf65 also belongs to the RF-1 family and has a similar domain architecture to ICT1; it is composed of the GGQ domain, followed by an unstructured C-terminal extension[17]. Except for an abundance of basic residues, there is no apparent sequence similarity in the C-terminal extension between ICT1 and C12orf65 (Supplementary Fig. 1), implying different interactions with ribosomes. As in the case for ICT1, genes homologous to *C12orf65* are found in most eukaryotes; however, in contrast to *ICT1*, they have not yet been detected in bacteria. The study of C12orf65 began with the identification of loss-of-function mutations in the nuclear gene, *C12orf65*, in two unrelated pedigrees that led to mitochondrial encephalomyopathy[18]. An analysis of fibroblasts from these patients confirmed that the mutations caused a decrease in the protein expression of all mtDNA genes, as well as in the levels of OXPHOS complexes[18–22]. Furthermore, overexpression of ICT1 in fibroblasts from patients lacking C12orf65 resulted in a 1.5-fold increase in cytochrome c oxidase (COX) activity; however, it is only 60% of that of wild type cells[18]. This suggests that C12orf65 has functions that are similar to and partially overlap with those of ICT1. Based on these findings alone, C12orf65 is regarded as another srRF. Nevertheless, in contrast to ICT1, release-factor assays using *E. coli* S30 fractions rich in ribosomes did not confirm that C12orf65 exhibited any detectable PTH activity in the presence or absence of any codon in the A-site of a ribosome.

As for fungi, homologous genes of *ICT1* and *C12orf65* have been found in the fission yeast *Schizosaccharomyces pombe* (*pth4* and *pth3*) and the budding yeast *Saccharomyces cerevisiae* (*PTH4* and *RSO55*, respectively) (Table 1). In this study, we refer to *RSO55 as PTH3* to avoid confusion between the species. There is a report on genetic experiments in *S. pombe* indicating that either of *pth4* and *pth3* is required for respiratory growth, and thus the two encoded proteins have partially overlapping functions in mitochondrial biogenesis[23]. However, none of the experiments

**Table 1 Summary of gene names of two stalled-ribosome rescue factors (srRFs) and one release factor (RF1), all of which have the GGQ motif, in mitochondria of four eukaryotes and *E. coli*.**

|  | srRF1 | srRF2 | RF1[a] |
|---|---|---|---|
| *S. cerevisiae* | PTH4 YOL114C | PTH3 RSO55 YLR281C | MRF1 YGL143C |
| *S. pombe* | pth4 SPAC589.11 | pth3 SPBC1105.18c | mrf1 SPAC2F7.17 |
| *H. sapiens* | ICT1[b] MRPL58 | C12orf65 | MTRF1L[c] MTRF1A |
| *E. coli* K-12 | arfB yaeJ | – | prfA |

[a]Most bacteria have two RFs, RF1 (*prfA*) and RF2 (*prfB*), whereas mitochondria from many eukaryotes have an RF1 ortholog, but no RF2 ortholog.
[b]Note that *ICT1* in *S. cerevisiae* is assigned to a different gene (*YLR099C*).
[c]In contrast to eukaryotic unicellular organisms, human cells have another GGQ-containing protein gene, *MTRF1*, of unknown functions.

clarified the primary role of the C12orf65 homolog, Pth3, as srRF. Although comprehensive proteomic analysis of mitochondrial proteins from *S. cerevisiae* confirmed that Pth4 is present in a ribosome-bound and in a free form in mitochondria[24,25], while Pth3 is localized in mitochondria[26], little is known about the physiological and biochemical functions of the two proteins in mitochondria.

This study not only focuses upon characterization of Pth4 and Pth3 from *S. cerevisiae* as srRFs, but also covers ribosome stalling caused by exogenous factors, namely, antibiotics that some types of microorganisms produce to inhibit the growth of others for survival. The binding of antibiotics to ribosomes may create a stalled-ribosome type that differs from a non-stop type because antibiotic-bound ribosomes are stalled somewhere on intact full-length mRNAs. Thus, the mRNA entry channels of the ribosomes remain occupied by mRNA, unless A-site mRNA cleavage occurs, as described later. This type of stalled ribosome is termed a "no-go" ribosome. ICT1/YaeJ do not appear to preferentially rescue this type of stalled ribosome because efficient PTH activity required the mRNA entry channel lacking mRNA[14,27]. Many of the antibiotics that bind to bacterial ribosomes bind to mitoribosomes, not eukaryotic cytoplasmic ribosomes, to impair cell viability and function through mitochondrial dysfunctions, and, in the case of clinical treatments, sometimes cause severe side effects[28–30].

In the present study, we demonstrated that the *S. cerevisiae* homolog of C12orf65, Pth3, preferentially rescued stalled mitoribosomes caused by various antibiotics that have different mechanisms of action even more efficiently than the homolog of ICT1, Pth4. Therefore, it appears that Pth3 is a mitochondrial srRF for antibiotic-bound ribosome stalling and plays a role in antibiotic resistance in fungi.

## Results

**Loss of *PTH4* and *PTH3* leads to a synthetic respiration defect under mild heat stress.** To assess the functions of the *S. cerevisiae* ICT1 and C12orf65 homologs, the Pth4 and Pth3 proteins, respectively, we constructed single-gene deletion mutants of *S. cerevisiae*, *pth4Δ* and *pth3Δ*, and a double-gene deletion mutant, *pth4Δpth3Δ*. Each gene was deleted by replacing its ORF with the *HIS3MX* module, an auxotrophic marker that complements *his-*, instead of any antibiotic markers, such as *KanMX6*, because antibiotics were used as translation inhibitors in experiments, as described below. We examined the growth of these mutants using a dilution spot assay on YPG agar plates containing glycerol as a non-fermentable carbon source. The growth of *S. cerevisiae* in YPG medium requires respiration and, thus, mitochondrial gene expression. At 30 °C, the *pth4Δ* and *pth3Δ* single mutants grew similarly to the wild type, while the *pth4Δpth3Δ* double mutant grew slightly slower than the wild type (Fig. 1a). These results showed that neither Pth4 nor Pth3 is indispensable for respiratory growth on YPG media under standard laboratory conditions. At 37 °C, as observed at 30 °C, the *pth4Δ* and *pth3Δ* single mutants grew as well as the wild type in YPG media, whereas the *pth4Δpth3Δ* double mutant appeared to be lethal in YPG but viable in glucose-containing media (YPD) (Fig. 1a). These results showed a synthetic lethal interaction or a negative genetic interaction between *PTH4* and *PTH3*, indicating that Pth4 and Pth3 have overlapping functions in mitochondria.

**Pth3 is particularly required for respiratory growth in the presence of bacteriostatic ribosome-binding antibiotics.** To clarify whether Pth4 and Pth3 are involved in the rescue of antibiotic-bound stalled mitoribosomes, we examined the effects on the growth of *pth4Δ*, *pth3Δ*, and *pth4Δpth3Δ* mutants on YPG

plates containing various types of antibiotics. Among antibiotics that bind to bacterial ribosomes, we selected nine that were reported to inhibit mitochondrial protein synthesis in yeast and/or humans (Supplementary Table 1). We initially confirmed that none of the antibiotics at the indicated concentrations exerted any apparent effects on the growth of the wild type or mutants on YPD plates, on which growth does not require mitochondrial gene expression for respiration (Supplementary Fig. 2). Furthermore, we established the adequate concentrations of each antibiotic needed to clarify differences in growth (Supplementary Fig. 3).

Spot assays on YPG plates showed that all of the antibiotics impaired growth more of the mutants than of the wild type (Fig. 1b). Comparisons of the growth of the mutants among the antibiotics tested revealed two patterns of inhibitory effects on growth that depended on the properties of the antibiotics, namely, bacteriostatic or bactericidal, rather than the mechanisms by which each antibiotic stalls ribosomes. In the presence of bacteriostatic antibiotics (tetracycline (Tc), oxytetracycline, doxycycline, azithromycin, and erythromycin), differences in effects on growth were observed among the three mutants. The order of decrease in growth rate appears to be *pth4Δpth3Δ* > *pth3Δ* > *pth4Δ*. As for chloramphenicol, little difference in effects on growth was observed between *pth4Δ* and the wild type. These results showed that Pth3 and, to a lesser extent, Pth4 were required for respiratory growth on YPG plates in the presence of the bacteriostatic antibiotics.

In contrast, in the presence of bactericidal antibiotics (paromomycin, tobramycin, and streptomycin), the deletion of *PTH4*, *PTH3*, or both equally impaired cell growth to a substantial extent (Fig. 1b). These results showed that Pth4 and Pth3 are both involved in efficient respiratory growth in the presence of bactericidal antibiotics.

However, recent experiments using bacteria indicated that bactericidal antibiotics, regardless of their mechanisms of action, induced cell death by a common mechanism involving the production of highly deleterious hydroxyl radicals[31,32]. In mammalian cells, similar bactericidal antibiotics have also been reported to induce the overproduction of reactive oxygen species (ROS) and mitochondrial dysfunction, leading to oxidative damage to DNA, proteins, and membrane lipids[33]. Thus, difficulties would be associated with assessing the proper functions of the two proteins as a rescue factor of stalled ribosomes in mitochondria if using cells grown in the presence of bactericidal antibiotics. In subsequent experiments, we used bacteriostatic antibiotics that inhibits mitochondrial translation to impair cell growth.

**Antibiotic susceptibility phenotype of the *pth4Δpth3Δ* mutant can be fully suppressed by plasmid-borne *PTH3*, but not *PTH4*.** To confirm a particular role of Pth3 in respiratory growth in the presence of antibiotics, we examined whether the severe slow growth phenotype of the double-gene deletion mutant was complemented by the plasmid-borne expression of either *PTH4* or *PTH3*. We transformed the *pth4Δpth3Δ* mutant with a derivative of the plasmid pRS316 containing the ORFs of *PTH4* or *PTH3* flanked with the native promoter and terminator, termed pPTH4 and pPTH3, respectively. Since pRS316 is a YCp (yeast centromeric plasmid) vector and, thus, the copy number is essentially one per cell, the protein expression level of the target gene in a transformed cell was assumed to be virtually identical to that in an untransformed wild-type cell. This plasmid also carries *URA3* as an auxotrophic marker gene. In the absence of antibiotics, the spot assay on synthetic medium plates lacking uracil and containing 3% glycerol (SG-ura) showed that the slow growth phenotype of the double-gene deletion mutant was well

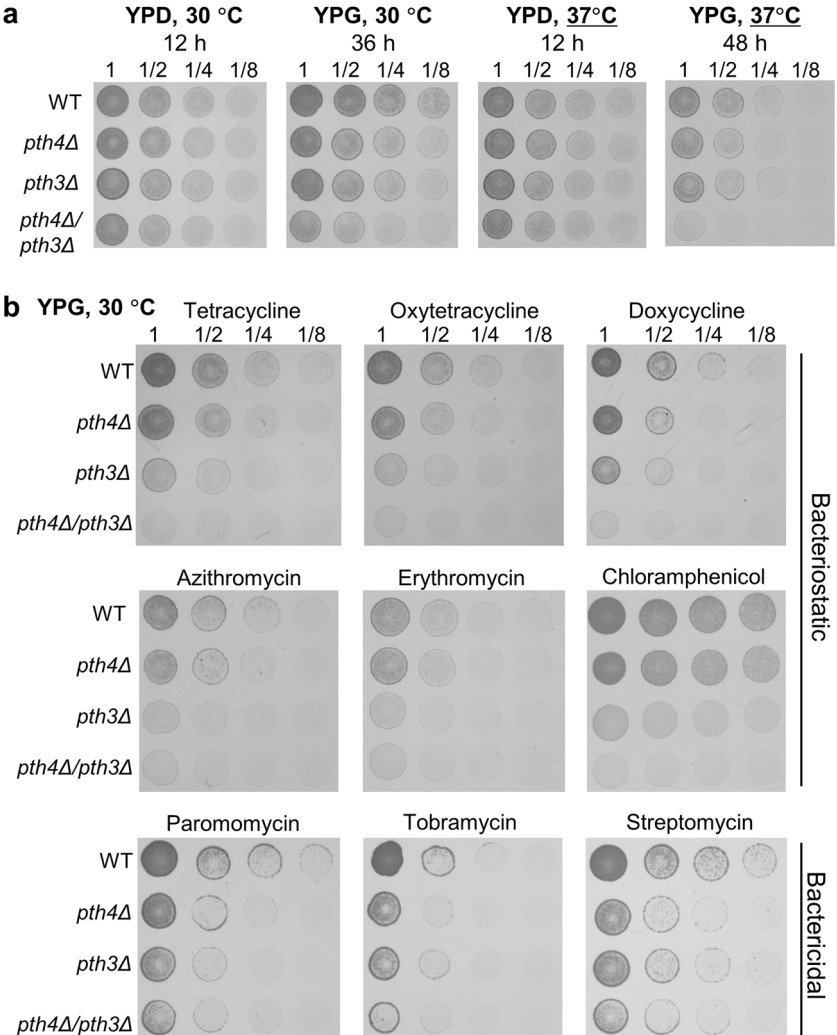

**Fig. 1 Susceptibility of *PTH4*- and/or *PTH3*-deficient mutants of *S. cerevisiae* to temperature and antibiotics. a** The wild type (WT), *pth4Δ*, *pth3Δ*, and *pth4Δpth3Δ* were serially diluted and then spotted onto YPD and YPG plates containing glucose and glycerol as a carbon source, respectively. Plates were incubated at 30 °C or 37 °C for the indicated hours. The results of all of the spotting assays in the present study are shown for one experiment representative of at least three independent experiments. **b** The four strains were serially diluted and then spotted onto YPG plates containing the following antibiotics, with the concentration shown in parentheses (μg/mL): tetracycline (100), oxytetracycline (200), doxycycline (100), azithromycin (75), erythromycin (20), chloramphenicol (1000), paromomycin (100), tobramycin (100), and streptomycin (1000). Plates were incubated at 30 °C for 48 h (tetracycline, oxytetracycline, and doxycycline) or 72 h (the others). Corresponding images of cells grown on YPD plates for 24 h are shown in Supplementary Fig. 2.

complemented by either the plasmid-borne *PTH4* or *PTH3* gene (Fig. 2 and Supplementary Fig. 4a). This result suggested that under standard laboratory conditions, one of Pth4 and Pth3 can rescue troubled mitoribosomes independently.

We examined whether the severely increased susceptibility to Tc, chloramphenicol, or azithromycin caused by the *pth4Δpth3Δ* mutant is suppressed by the expression of *PTH4* or *PTH3*. The spot assay on SG-ura plates in the presence of each antibiotic demonstrated that the antibiotic susceptibility phenotype of the *pth4Δpth3Δ* mutant was fully suppressed by plasmid-borne *PTH3*, whereas it was only partially suppressed by plasmid-borne *PTH4* (Fig. 2). Additionally, to check if the partial suppression was improved by more increased gene expression of *PTH4*, we replaced the native promoter with a strong constitutive TEF1 promoter in pPTH4 (Supplementary Fig. 4b). However, it resulted in a decrease rather than an increase in the suppression efficiency using the TEF1 promoter than the native promoter. It was also found that overexpression of Pth3 considerably impaired growth even in the absence of Tc (Supplementary Fig. 4b). It is possible

that an excessive amount of either of srRFs causes its unexpected binding to normal translating mitoribosomes, preventing normal elongation or termination. These results revealed that Pth3 plays a particular role in antibiotic resistance, suggesting the presence of specific mitoribosome stalling caused by the antibiotics that is only rescued by Pth3.

Finally, to confirm that the PTH activities of Pth4 and Pth3 directly contributed to the suppression of the antibiotic susceptibility phenotypes of the *pth4Δpth3Δ* mutant, we introduced mutations into plasmids such that the GGQ motif residues were changed to VAQ. These mutations decreased the PTH activities of YaeJ and ICT1 in the in vitro translation, as previously described[9,34]. The spot assay showed that none of the antibiotic susceptibility phenotypes were suppressed by plasmid-borne *PTH3* or *PTH4* mutant having VAQ residues (Fig. 2). These results showed that the GGQ motif residues are required for the functions of Pth4 and Pth3, indicating that Pth3 and Pth4 hydrolyze peptidyl-tRNAs on ribosomes in vivo with the GGQ motif residues as the catalytic site.

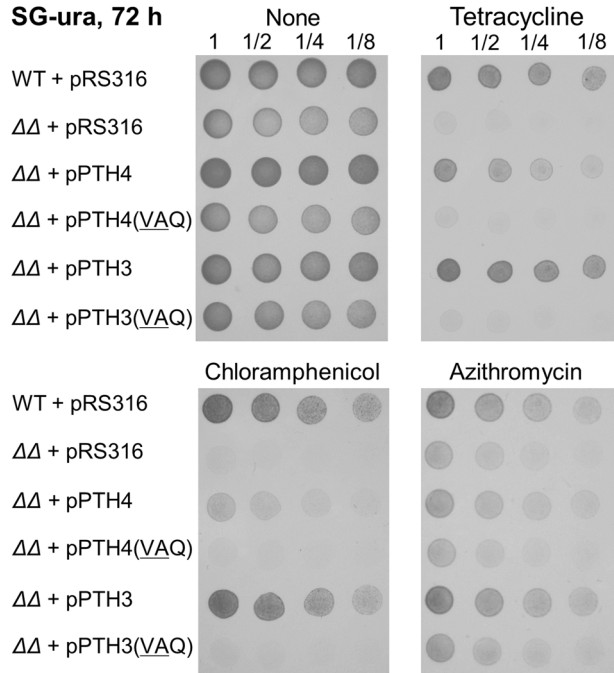

**SG-ura, 72 h**

**Fig. 2 Suppression of the antibiotic susceptibility phenotype of the double-gene deletion mutant by plasmid-borne *PTH3* or *PTH4*.** The *pth4Δpth3Δ* mutant (represented by ΔΔ) was transformed with the plasmid pRS316 harboring a genomic region including PTH4 and PTH3, termed pPTH4 and pPTH3, and the resultant strains were labeled ΔΔ+pPTH4 and ΔΔ+pPTH3, respectively. Each strain was diluted and spotted onto SG-ura plates containing no antibiotic, 100 μg/mL tetracycline, 1000 μg/mL chloramphenicol or 75 μg/mL azithromycin. As a control, *pth4Δpth3Δ* as well as the wild type was transformed with an empty plasmid, pRS316. In addition, *pth4Δpth3Δ* was transformed with the plasmid in which the GGQ motif residues were changed to VAQ in pPTH4 and pPTH3, termed pPTH4 (VAQ) and pPTH3(VAQ), respectively. Corresponding images of cells grown on SC-ura plates for 12 h are shown in Supplementary Fig. 4a.

**Unstressed mitochondria require the functioning of Pth4 and Pth3.** To investigate whether the deletion of *PTH4* or *PTH3* exerted any effects on mitochondrial properties in the absence or presence of Tc, we examined the mitochondrial mass and membrane potential of the mutants using flow cytometry (FCM).

First, each growth of the mutants was examined in liquid YPG media by assessing optical density at 600 nm at three time points. In the absence of Tc, growth rates were essentially the same between the two single-gene deletion mutants and the wild type, while a small decrease in growth of the *pth4Δpth3Δ* mutant compared to that of the wild type was only observed in the exponential growth phase (Fig. 3a and Supplementary Fig. 5a). These results were basically consistent with those of the spot assay on solid agar plates. In the presence of Tc, the concentration in liquid medium was increased to 500 μg/mL so that differences of growth between the mutants and the wild type could be clarified (Supplementary Fig. 5b). As observed in the spot assay, growth of the *pth3Δ* mutant was slower than that of the wild type, whereas that of the *pth4Δpth3Δ* mutant was severely impaired (Fig. 3a and Supplementary Fig. 5a). However, growth of the *pth4Δ* mutant was virtually identical to that the wild type. This result differed from that obtained using YPG plates containing Tc on which *pth4Δ* grew more slowly than the wild type. Thus, the Tc sensitivity of *pth4Δ* appears to be less prominent in liquid medium.

In the FCM analysis, mitochondrial mass and membrane potential in cells grown in liquid YPG media for 48 h were measured using the uptake of MitoTracker Green FM and Red CMXRos, respectively. The uptake of the former dye is dependent on mass, but not mitochondrial potential, whereas that of the latter has the opposite characteristics. Relative to the major peak position observed in the wild type, rightward peak shifts in the fluorescent detection of mitochondrial mass were observed in *pth4Δ* and *pth3Δ*, whereas no marked changes were noted in those of membrane potential in the three mutants (Fig. 3b). These results indicated an increase in the number of mutant cells that have larger mitochondrial mass with normal membrane potential compared with those of the wild type. This increase in mitochondrial mass has been observed in mitochondrial myopathies and is considered to partly compensate for the reduced function of the respiratory chain by maintaining overall ATP production[35,36]. Thus, the putative reduced function of mitochondria in the mutants appeared to be complemented by the increased mitochondrial mass. These results suggest that the lack of *PTH4* or *PTH3* places a stress on mitochondrial functions, even under standard laboratory conditions; in other words, unstressed mitochondria require the functioning of both Pth4 and Pth3.

Furthermore, in the presence of Tc, the FCM analysis of yeast cells grown for 48 h showed few differences in mitochondrial mass and membrane potential between the wild type and the mutants (Fig. 3b). In addition, the FCM analysis using the oxidation-sensitive probe 2',7'-dichlorofluorescein diacetate (DCFH-DA) showed no substantial differences in ROS accumulation between the mutants and the wild type; however, a slight increase was observed over that in the absence of Tc (Supplementary Fig. 6a). We also examined whether the growth impairment of the mutants in the presence of Tc is due to growth arrest or loss of cellular viability. The mutants were grown in YPG liquid media containing Tc, paromomycin, or azithromycin for 1 h, and were washed with PBS buffer to remove each antibiotic. The samples were spotted on YPG plates in the absence of the antibiotics, showing that the mutants grew as well as the wild type (Supplementary Fig. 6b). Combined with the results of the FCM analysis, these results suggest that the antibiotics at each sublethal concentration arrest cell growth of the mutants rather than reduce cell viability.

Taken together, these findings indicate that Tc inhibits the mutant-specific increase of mitochondrial mass that was observed under non-stressful conditions, and it does not damage mitochondrial functions in the mutants enough to activate cellar stress responses. Yeast cells that have these properties were used in subsequent experiments.

**Expression of all mtDNA protein-coding genes appears to be decreased only in *pth3Δ* in the presence of Tc.** To examine the effects of the deletion of *PTH4* or *PTH3* on the protein synthesis of mtDNA protein-coding genes in mitochondria, we labeled mitochondrial translation products in organello with [35S] methionine. Consistent with the findings reported by Meisinger et al.[37], we obtained a mitochondrion-rich fraction from the wild type and the mutants grown in YPG media in the absence or presence of Tc for 24 h. Fresh mitochondria were incubated for 25 min in the presence of [35S]methionine. Mitochondria were collected by centrifugation, washed, and then directly analyzed by SDS-PAGE. Gels were imaged by imaging plate autoradiography (Fig. 4a and Supplementary Fig. 7a). All eight proteins encoded by *S. cerevisiae* mtDNA were identified as described previously[38–40]. In the absence of Tc, seven protein products were virtually identical among the three strains, although the Var1 levels were slightly higher in *pth3Δ* than in the wild type or *pth4Δ* (Fig. 4a and Supplementary Fig. 7b). Var1, which is the only hydrophilic protein, serves as a component of the small ribosomal subunit of

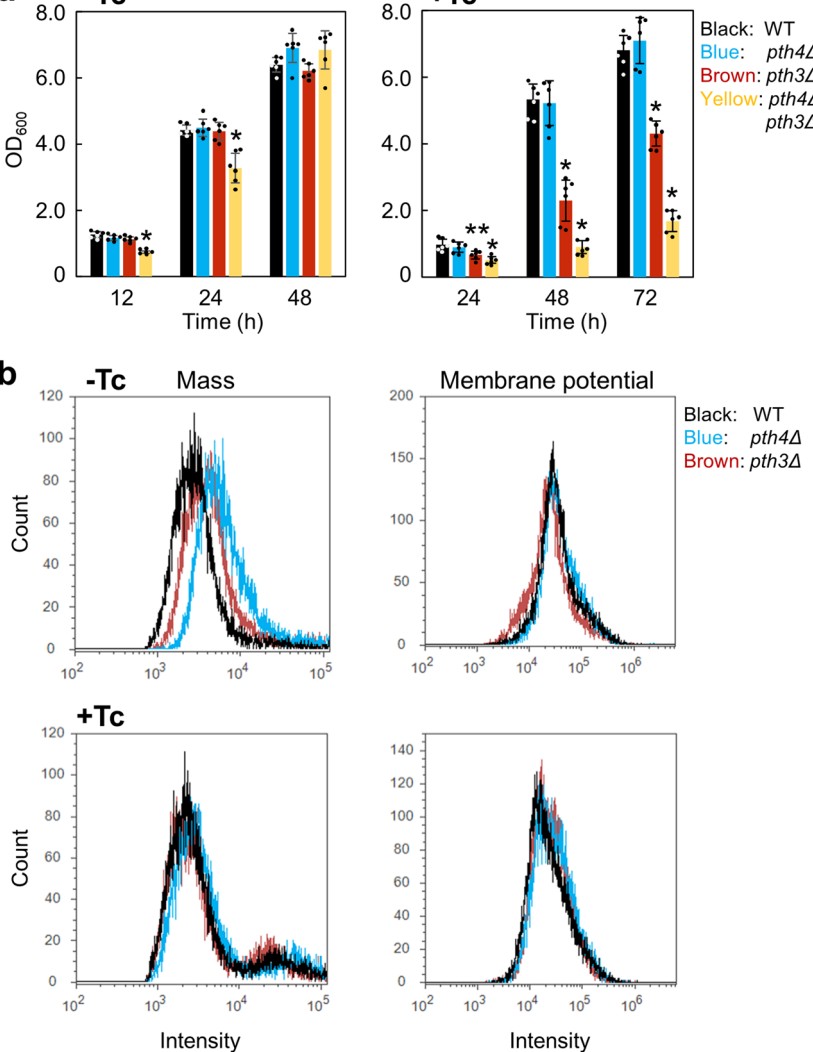

**Fig. 3 Effects of the deletion of *PTH4* or *PTH3* on mitochondrial properties. a** The three mutants and the wild type were inoculated into liquid YPG media in the absence or presence of 500 µg/mL tetracycline (Tc) with a starting $OD_{600}$ of 0.01. Growth was monitored by measuring the $OD_{600}$ values at the indicated time points at 30 °C. Data are shown as the mean ± standard deviation of six independent experiments. Asterisks indicate significant difference from the wild type (Student's t test, *$P < 0.001$ and **$P < 0.01$). **b** Mitochondrial mass (left) and membrane potential (right) in the wild type (WT), *pth4Δ*, *pth3Δ*, and *pth4Δpth3Δ* grown in liquid YPG media for 48 h were measured at room temperature (about 25 °C) by FCM using MitoTracker Green FM and MitoTracker Red CMXRos, respectively, in the absence or presence of 500 µg/mL tetracycline (Tc).

mitoribosomes, and in many other species, its homologous proteins are encoded by nuclear DNAs. The small increase in the Var1 levels may be indicative of increase in the number of mitoribosomes.

In contrast, in the presence of Tc, all eight protein products appeared to be decreased in *pth3Δ* only (Fig. 4a and Supplementary Fig. 7b). These results in the absence or presence of Tc were consistent with those from comparisons of growth rates in liquid YPG media among the three strains (Fig. 3a). Since Tc binds to mitoribosomes to inhibit translation, these results indicated that the lack of Pth3, but not Pth4, prevented mitochondrial protein synthesis, suggesting that Pth3 exclusively rescues mitoribosomes stalled by Tc. However, the possibility cannot be excluded that the reduction rates in protein products differ from those in vivo.

We performed quantitative reverse transcription PCR (qPCR) to confirm whether the deletion of *PTH4* or *PTH3* altered the transcriptional levels of mtDNA protein-coding genes in the absence and presence of Tc. Because suitable specific primers were not able

to be designed for *VAR1*, which is highly AT-rich, the mRNA levels of seven genes were examined. Since *VAR1* and *ATP9* are co-transcribed in mitochondria from *S. cerevisiae*, changes in the mRNA expression of *VAR1*, if ever, may depend on those in *ATP9*. Regarding *pth4Δ* and *pth3Δ*, the ratio of mutant to wild-type mRNA levels of all genes was in the range from 0.75 to 1.34 factors in the absence of Tc and in the range of 0.82–1.28 factors in the presence of Tc (Fig. 4b). These results showed that the deletion of either gene had little effect on the mRNA expression of mtDNA protein-coding genes. Additionally, we confirmed if Tc affected gene expression of *PTH4* or *PTH3* in the wild type. No substantial changes were detected in expression of either genes (Supplementary Fig. 7c).

Thus, these results confirmed that a significant decrease in protein products in *pth3Δ* in the presence of Tc basically occurs at the post-transcriptional or translational level.

**Pth3 and Pth4 exhibit PTH activity in the in vitro translation system from *E. coli*.** The human homolog of Pth4, ICT1, was

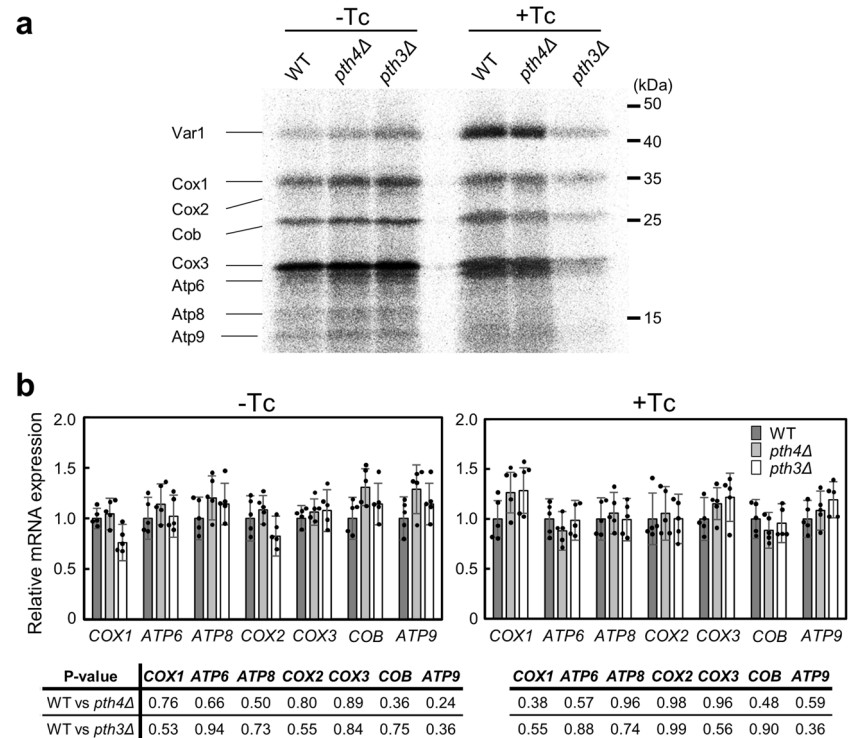

**Fig. 4 Effects of the deletion of *PTH4* or *PTH3* on mitochondrial translation and transcription in the absence or presence of tetracycline. a** Mitochondrial translation products synthesized in organelles from the wild type (WT), *pth4*Δ, and *pth3*Δ grown in YPG media for 24 h in the absence or presence of 500 μg/mL tetracycline (Tc). Fresh mitochondria isolated from each strain were incubated with [$^{35}$S]methionine in translational buffer. Mitochondrial lysates were analyzed by 16% SDS-PAGE and autoradiography. The corresponding Coomassie blue-stained gels are shown in Supplementary Fig. 7a. Quantitative comparison of mitochondrial translation products is shown in Supplementary Fig. 7b. **b** qPCR data of all mtDNA protein-coding genes, except for *VAR1*, from total RNAs extracted from the three strains in the absence or presence of 500 μg/mL Tc. Data are shown as the mean ± standard deviation of five independent experiments. *P* values are shown in the tables. There were no statistical significant differences between the wild type and each mutant in the absence or presence of Tc (Student's *t* test, *p* > 0.1). The mRNA expression of *VAR1* was described in the text.

shown to exhibit PTH activity toward non-stop mRNA similar to the *E. coli* homolog YaeJ, using a cell-free protein synthesis system reconstituted from *E. coli* purified components (PURE-system)[11,14]. We investigated whether Pth3 and Pth4 exhibit PTH activity towards these stalled ribosomes using the same in vitro translation system. Using a non-stop template containing the *crp* gene lacking a stop codon (Supplementary Fig. 8a), an in vitro translation reaction was performed for 15 min to produce stalled ribosomes with peptidyl-tRNAs (CRP-tRNAs). Various concentrations of the recombinant Pth4 or Pth3 protein were added to these ribosomes. According to previous findings on ICT1 and YaeJ[11], a His-tag was added to the C terminus of Pth4, whereas that tag was added to the N terminus of Pth3 because the N-terminal tag exerted no significant effect on PTH activity (Supplementary Fig. 8b). In vitro translation products were analyzed using the NuPAGE Bis-Tris electrophoresis system. Products labeled fluorescently with tRNA$^{Lys}$ charged with a fluorescently labeled lysine were detected and quantified using ImageQuant LAS 4010 (Fig. 5a). The ratio of the band intensities of released peptides to that of peptidyl-tRNA plus those of released peptides was defined as relative PTH activity for stalled ribosomes. Thus, 100% relative activity indicated that peptidyl-tRNAs on stalled ribosomes were fully hydrolyzed.

Figure 5a shows the concentration dependence of the PTH activities of Pth4, Pth3 for non-stop ribosomes. Pth4 exhibited PTH activity at a similar level to that of ICT1 and YaeJ, as previously described[11]. Pth4 appears to serve as a rescue factor for non-stop ribosomes in *S. cerevisiae* mitochondria. In addition, the results obtained demonstrated that Pth3 exhibited PTH activity

for non-stop mRNA that was weaker than that of Pth4. To confirm that the GGQ motif residues of Pth3 and Pth4 contributed to PTH activity in this heterogeneous translation system, we changed the GGQ residues to VAQ using site-directed mutagenesis. The mutation in both mutant proteins resulted in a significant decrease in PTH activity, which demonstrated the need for GGQ residues for PTH activity (Supplementary Fig. 8c).

**Pth3 exhibits a similar level of PTH activity between non-stop and no-go ribosomes from *E. coli*.** We attempted to test whether Pth3 or Pth4 rescues no-go ribosomes caused by an antibiotic using the in vitro translation system. However, an antibiotic works at miscellaneous sites on intact mRNA, so that it would produce heterogeneous no-go ribosomes, leading to the lack of a discrete band on the gel. Instead, we constructed a DNA template in which the stop codon of the *crp* gene was followed by 33 nucleotides; we named mRNA derived from the template no-go mRNA (Supplementary Fig. 8a). Since RFs were omitted from the translation mix of the PURExpress ΔRF123 kit, ribosomes stalled at the stop codon of no-go mRNA. According to Feaga et al.[14], the mRNA entry channel of a ribosome should be sufficiently occupied by 33 nucleotides. We confirmed that the band indicating peptidyl-tRNAs on no-go ribosomes was undetectable when RF was added to the solution (Supplementary Fig. 8d).

Next, we found that ≈60% PTH activity of Pth4 for non-stop and no-go ribosomes required 0.2 μM and 1.5 μM, respectively (Fig. 5b). The result showed that as expected in experiments on ICT1 and YaeJ[14,27], the PTH activity of Pth4 was weaker towards

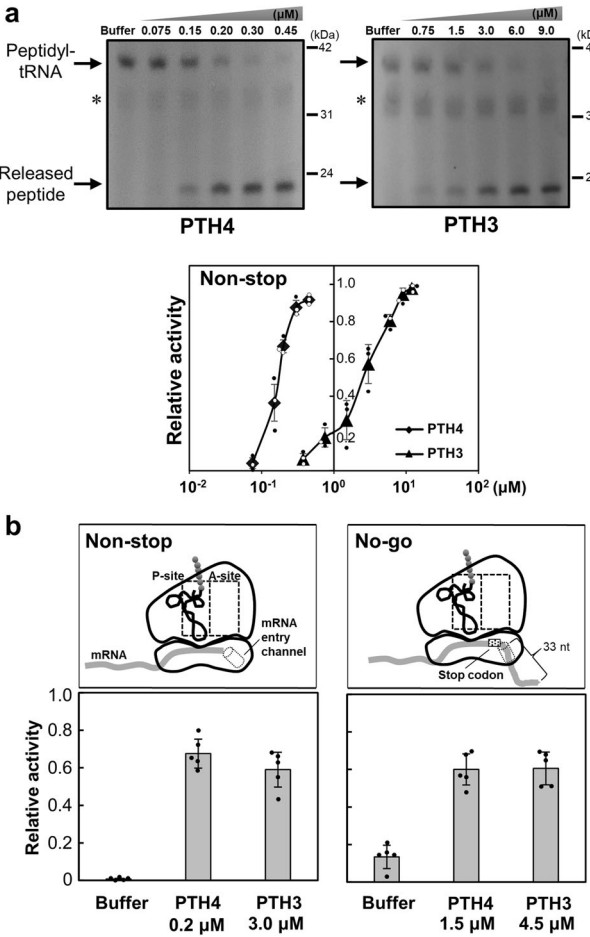

## Discussion

The present study characterized the functions of Pth3 and Pth4 as mitochondrial srRFs in *S. cerevisiae*. Under standard laboratory conditions with a non-fermentable carbon source, which requires respiration and, thus, mitochondrial gene expression, the deletion of either *PTH4* or *PTH3* had no apparent effects on respiratory growth, while that of both genes had a small effect. According to the FCM analysis, mitochondrial mass in each gene deletion mutant increased possibly through mitochondrial dynamics in which mitochondria undergo coordinated cycles of fission and fusion[41]. Conversely, this result implies that both srRFs play important roles in maintaining unstressed or normal mitochondria. Under mild heat stress, *PTH3* and *PTH4* were found to have a synthetic lethal genetic interaction, suggesting that either Pth3 or Pth4 is required for respiratory growth under this stress. This finding shows functional redundancy between Pth3 and Pth4 as srRFs in yeast mitochondria.

In contrast, the inhibition of mitochondrial translation by ribosome-targeting antibiotics highlighted functional differences between the two srRFs. The deletion of *PTH3* more significantly impaired respiratory growth in the presence of bacteriostatic antibiotics, such as Tcs, macrolides, and chloramphenicol, than that of *PTH4*. The increased susceptibility to Tc, chloramphenicol or azithromycin caused by the double-gene deletion on non-fermentable plates was fully suppressed by plasmid-borne expression of *PTH3* from the native promoter, whereas it was only partially suppressed by that of *PTH4* form the native promoter or a strong TEF1 promoter. These results suggest the presence of some specific ribosome stalling caused by the antibiotics that can only be recused by Pth3. Additionally, radioactive labeling experiments in organello and qPCR experiments for the quantification of mRNA levels suggested that in the presence of Tc, the deletion of *PTH3*, but not *PTH4*, decreased the mitochondrial translation of all mtDNA-encoded protein genes. It is plausible that the lack of Pth3 leaves Tc-bound mitoribosomes stalled on mRNAs from all eight genes, which may not be efficiently rescued by Pth4. These results indicate that Pth3 can function as a rescue factor for antibiotic-dependent stalled mitoribosomes. In nature, antibiotics are produced by one microorganism to kill or inhibit the growth of competitor microorganisms. Fungi may be exposed to ribosome-binding antibiotics in some environments. In order for fungi to survive in these hostile environments, it is highly advantageous for them to have a special srRF in mitochondria that efficiently rescues antibiotic-stalled mitoribosomes. Therefore, it follows that Pth3 plays a role in antibiotic resistance in fungi.

It is important to note that various types of antibiotics that have different modes of action exerted significant effects on the respiratory growth of *pth3Δ*. For example, Tc primarily binds to rRNA in the small ribosomal subunit and blocks the entry of aminoacyl-tRNA into the A-site of the ribosome during translation[42]. Em binds to the nascent peptide exit tunnel of the large ribosomal subunit to impede the passage of newly synthesized polypeptides in a context-specific manner[43]. Based on similarities in stalled ribosomes caused by the various antibiotics used in this study, it appears that the action of Pth3 only requires the A-site to be left empty, at least intermittently, in the stalled ribosome in which the mRNA entry channel is occupied by intact mRNA; this stalled ribosome represents a no-go type. Accordingly, the rescue target of Pth3 may be extended to no-go ribosomes stalled for reasons other than antibiotics, for example, the lack of specific tRNAs, rare codon clusters, and the misfolding of a nascent peptide within the inner membrane, which is suggested to frequently occur during the co-translational process[44]. In bacterial translation, the endonucleolytic cleavage of a problematic mRNA in the A-site by RelE or other factors often converts no-go into

**Fig. 5 PTH activities of Pth4 and Pth3 toward non-stop or no-go ribosomes using the *E. coli*-based reconstituted translation system. a** In vitro translation of the non-stop template with recombinant His-tagged Pth4 and Pth3 proteins. Each recombinant protein was added to the solution in which a 15-min in vitro translation reaction had been performed using the non-stop templates. The resulting mixture, incubated for 5 min, was analyzed by NuPAGE, shown in the left panel. Each gel was visualized using a laser-based fluorescent gel scanner. Released peptides and peptidyl-tRNA indicate Crp proteins and Crp-tRNA, respectively. The final concentration of each recombinant protein is shown in each lane. An asterisk indicates a background band (≈30 kDa) that appears without the addition of template DNA or mRNA, as previously shown in the manufacturer's technical note. The right panel shows the extent of peptidyl-tRNA hydrolysis (relative PTH activity) given as the ratio of the band intensity of released peptides to that of peptidyl-tRNA plus that of released peptides at the protein concentrations indicated. Data are shown as the mean ± standard deviation of three independent experiments. **b** Comparison of concentrations between Pth4 and Pth3 proteins required for similar PTH activities toward non-stop or no-go ribosomes. Each upper panels show a schematic drawing of a ribosome stalled on non-stop or no-go mRNA. The lower panels show that ≈60% PTH activities for non-stop and no-go ribosomes require the indicated concentrations of Pth4 or Pth3. Data are shown as the mean ± standard deviation of five independent experiments. An example of gel images is shown in Supplementary Fig. 8e.

no-go mRNAs than non-stop mRNAs. In contrast, ≈60% PTH activity of Pth3 for non-stop and no-go ribosomes required similar concentrations (3.0 μM and 4.5 μM, respectively). These results suggested that the PTH activity of Pth3 did not depend on whether the mRNA entry channel was occupied by mRNA, at least in the in vitro translation.

no-stop types of stalled ribosomes[45]. No information is currently available on this type of mRNA cleavage in mitochondria; however, the RNase II like protein, Dss1, reportedly bound to the large subunit of yeast mitoribosomes[46,47].

This series of findings is basically supported by those of the in vitro translation experiment using the E. coli-based reconstituted translation system. The results obtained showed that Pth4 exhibited a similar level of PTH activity to that of the E. coli counterpart YaeJ, and preferred the rescue of non-stop to no-go ribosomes. Similar results were observed for the human counterpart, ICT1[14,17]. Based on the conservation of the key amino acid sequence among the three proteins from different species (Supplementary Fig. 1), the rescue mechanism for non-stop ribosomes, which is mediated by YaeJ/ICT1/Pth4 homolog subfamily proteins, appears to be common among E. coli, human mitochondria, and yeast mitochondria. In addition, the present results showed that the PTH activity of Pth3 was similar between the two types of stalled ribosomes. This result is consistent with the suggestion that Pth3 equally enters into the A-site of stalled ribosomes with or without the presence of mRNA in the mRNA entry channel. However, the confirmation of Pth3 activity levels requires further experiments in a homogeneous system.

The present study suggested that the human Pth3 homolog, C12orf65, is the second srRF in human mitochondria. Since the knockdown of C12orf65 or ICT1 in human cells impaired mitochondrial translation and cell viability[10,17,18], C12orf65 and ICT1 appear to have a narrower range of overlapping functions than their counterparts from S. cerevisiae. Since human cells are apparently not attacked by such antibiotics as used in this study, except in the case of therapeutic use, C12orf65 may rescue the no-go types of ribosomes stalled for reasons other than antibiotics, as described above, even more efficiently than ICT1. It would be interesting to examine whether C12orf65 actually rescues mitoribosomes stalled by antibiotics in human cells and particularly cells from specific tissues affected by the associated side effects. The composition and structure of mitoribosomes as well as factors involved in translation, such as translation activators, have been shown to differ significantly between yeast and human mitochondria[8]. To advance research on srRFs, a deeper understanding is needed of mitochondria-specific ribosome stalling, which appears to differ among organisms.

## Methods

**Chemicals**. Antibiotics were from Fujifilm Wako (tetracycline hydrochloride, streptomycin sulfate, and chloramphenicol), Sigma-Aldrich (oxytetracycline hydrochloride, doxycycline hydrochloride, paromomycin sulfate, and tobramycin), and Tokyo Chemical Industry (azithromycin dehydrate and erythromycin).

**Strains, plasmids, and media**. Budding yeast strains and plasmids used in this study are listed in Supplementary Tables 2 and 3, respectively. Oligonucleotide primers used in this study are listed in Supplementary Table 4.

Cells were grown in YPD medium [1% (w/v) bacto-yeast extract, 2% (w/v) bacto-peptone, and 2% (w/v) glucose] or YPG medium [1% (w/v) bacto-yeast extract, 2% (w/v) bacto-peptone, and 3% (v/v) glycerol] in the absence or presence of each antibiotic. Synthetic medium containing 2% glucose (SC) or 3% glycerol (SG) was prepared as described previously by Hanscho et al.[48].

**Construction of gene deletions in S. cerevisiae**. To generate the pth4Δ (yol114cΔ)::HIS3 or pth3Δ (yolr281cΔ)::HIS3 strain, the entire ORF of PTH4 (YOL114C) or PTH3 (YLR281C) was replaced in a wild-type haploid by the His3MX6 cassette using a PCR-based gene targeting method[49] and the plasmid pFA6a-His3MX6[50]. The pth4Δ::HIS3 pth3Δ::HIS3 double mutant strain was generated by a genetic cross between single null mutants with the opposite mating type. Tetrad spores were separated under a microscope (CX31, Olympus, Japan) using a micromanipulator (Narishige, Japan). All gene deletions or integrations were confirmed by diagnostic PCR.

**Construction of plasmids**. The plasmids pPTH4 and pPTH3 were constructed as follows: The PTH4 or PTH3 ORF flanked with 500 bp of the 5′ UTR and 300 bp of the 3′ UTR was amplified by PCR using the DNA polymerase KOD-Plus-Neo (TOYOBO) and primers pRS316 114 C genomic (F) and pRS316 114 C genomic (R), and primers pRS316 281 C genomic (F) and pRS316 281 C genomic (R) from the genomic DNA of wild-type yeast cells, respectively. Each of the products was ligated into the Xho I/Xba I sites of pRS316 using the In-Fusion cloning kit (Takara). The pPTH4(VAQ) and pPTH3(VAQ) plasmids in which the GGQ motif residues (Gly80–Gly81–Gln82 and Gly49–Gly50–Gln51) (Supplementary Fig. 1) were changed to Val–Ala–Gln in pPTH4 and pPTH3, respectively, were prepared by site directed mutagenesis.

**Antibiotic susceptibility**. Serial dilutions of overnight cultures grown in YPD and SC-ura liquid media for ≈17 h were spotted onto YPG and SG-ura agar plates as well as YPD and SC-ura agar plates (controls), respectively, in the absence or presence of each antibiotic, and incubated at 30 °C. The range of effective concentrations differed depending on the type of antibiotic, and a series of dilution spot assays were performed at least at three different concentrations of each antibiotic (Supplementary Fig. 3).

**Fluorescent detection of mitochondrial membrane potential, mitochondrial mass, and ROS accumulation**. Yeast cells were grown in liquid YPG medium in the absence or presence of 500 µg/mL Tc in a baffled flask at 30 °C with a shaking incubator (130 rpm) for 24 or 48 h. MitoTracker Red CMXRos and MitoTracker Green FM (Thermo Fisher Scientific) were used to assess mitochondrial membrane potential and mass, respectively. Cells were harvested by centrifugation and resuspended in 100 µL of SG medium (3% glycerol). 1 µL of 1 mM MitoTracker red or green was added to each sample, and samples were incubated at room temperature for 15 min and then diluted with 3 mL of PBS. Cells were analyzed by FCM using the Attune acoustic focusing cytometer (Thermo Fisher Scientific). A total of 20,000 cells per sample were measured in each experiment. The flow cytometry gating strategy is shown in Supplementary Fig. 9.

Intracellular ROS accumulation was measured using the fluorescent probe 2′,7′-dichlorofluorescein diacetate (DCFH-DA) (Sigma-Aldrich). Cells were harvested by centrifugation and resuspended in 1 mL of PBS. DCFH-DA was added to each sample to a final concentration of 10 µM and samples were incubated for 2 h in the dark at 30 °C. As a positive control, $H_2O_2$ was also added to the sample of the wild type to a final concentration of 10 mM. The following procedure is described above.

**Analysis of mitochondrial translational products**. Mitochondria were prepared from yeast cells grown as described previously by Herrmann et al.[37]. Briefly, cells of each strain were collected by centrifugation and spheroplasts were obtained by digestion with Zymolyase 20T (Nacalai Tesque). Spheroplasts were homogenized by 15 strokes of a Potter–Elvehjem tissue grinder and mitochondria were isolated by differential centrifugation. After each isolated mitochondrial pellet was mixed with 1 mL of SH buffer, protein concentration (mg/mL) was determined with the Bradford assay (Bio-Rad). Each sample was diluted to a final concentration of 10 mg/mL in SH buffer, which contained 150 µg/mL cycloheximide (Nacalai Tesque) that blocks cytoplasmic translation for confirmation. Organelle labeling was performed as described previously by Herrmann et al.[51]. In total, 20 µL of each of the samples was mixed with 40 µL of the translation buffer as described. After the addition of [$^{35}$S]methionine (Perkin Elmer), the resultant solution was incubated at 30 °C for 25 min. To stop labeling, 10 µL of 200 mM nonlabel methionine was added. After an incubation for 5 min, 1 mL of SH buffer was added to the resultant mixture and mitochondria were then collected by centrifugation. The resultant mitochondrial pellets were resuspended with 50 µL of SDS sample buffer. All samples were directly analyzed by 16% SDS-PAGE. Radiolabeled mitochondrial protein products were visualized by exposure to an imaging plate (BAS-IP MS 2040 E, GE Healthcare) on the fluorescent image analyzer FLA-3000 (GE Healthcare). Uncropped gel images are presented in Supplementary Fig. 10.

**Quantitative reverse-transcription PCR (qPCR)**. RNA was extracted from cells of each strain using the PureLink RNA Mini Kit (Invitrogen), which includes a treatment step with DNase. The RNA integrity of the RNA samples was assessed by formaldehyde agarose gel electrophoresis (Supplementary Fig. 11). 1 ng of total RNA was used as a template. qPCR was performed using the One Step TB Green PrimeScript RT-PCR Kit II (Takara) on a LightCycler 480 system II (Roche Applied Science). ACT1 was used as an internal reference gene in accordance with qPCR experiments for the expression analysis of mtDNA genes. The qPCR results were analyzed using a LightCycler 480 system software based on ΔΔCt method. Sequences for the primers are shown in Supplementary Table 5.

**Protein expression and purification of recombinant His-tagged proteins**. The DNA sequences encoding the PTH4 or PTH3 genes were amplified by PCR using the genome of S. cerevisiae strain BY4742 provided by Open BioSystems as templates. The primers used in the present study are listed in Supplementary Table 4. The recombinant Pth4 and Pth3 proteins were designed as truncated proteins lacking 37 and nine N-terminal residues, respectively, which were predicted to function as an import signal into mitochondria from fungi by the MitoFates server[52]. In addition, C-terminal and N-terminal His₆-tags were added to the Pth4 and Pth3 proteins,

respectively, as described in detail in the results section. To obtain recombinant Pth4 and Pth3 proteins, amplified DNA fragments were digested with *Nde* I (New England Biolabs) and *Xho* I, and with *Nde* I and *BamH* I, and then cloned into the expression vectors pET26b and pET15b (Novagen), respectively, using the In-Fusion HD cloning kit. Vectors were transformed into *E. coli* BL21-CodonPlus(DE3)-*RIPL* cells (Novagen). Protein expression and purification were subsequently performed as described previously[11]. Protein concentrations were measured using the 2-D Quant kit (GE Healthcare).

**PTH assay**. The PURExpress delta RF123 kit (New England Biolabs), based on PUREsystem technology, was used for in vitro translation experiments with the FluoroTect Green$_{Lys}$ in vitro translation labeling system (Promega). In this kit, RF1, RF2, and RF3 are omitted from the translation mix and are supplied separately. A non-stop template with TA at the 3′ end was prepared from the *crp* gene, while a no-go template with a stop codon followed by 33 nt was prepared from the *crp* gene (Supplementary Fig. 8a). Using each template, an in vitro translation reaction was performed at 37 °C for 15 min to produce stalled ribosomes with peptidyl-tRNAs, to which the in vitro translation mixture containing various concentrations of purified recombinant Pth4 or Pth3 proteins was directly added. The resulting mixture was incubated for 5 min and then analyzed by NuPAGE using the NuPAGE Bis–Tris electrophoresis system (Invitrogen). Direct "in-gel" detection of the proteins containing fluorescently labeled lysine residues was accomplished using ImageQuant LAS 4010. Band intensities corresponding to the CRP protein or CRP-tRNA were quantified using ImageQuant software, and the efficiency of peptidyl-tRNA hydrolysis was calculated as the ratio of the CRP protein to CRP-tRNA plus the CRP protein. The ribosome concentration in the reaction solution of PURExpress was 2.4 μM. Uncropped gel images are presented in Supplementary Fig. 10.

**Statistics and reproducibility**. Results of the spotting assays or FCM measurements are shown for one experiment representative of at least three independent experiments. Specific information on the number of replicates and statistical analysis is included in each figure legend.

**Reporting summary**. Further information on research design is available in the Nature Research Reporting Summary linked to this article.

## Data availability

The datasets generated during and/or analysed during the current study are available from the corresponding author on reasonable request. The source data underlying plots shown in main figures are provided in Supplementary Data 1.

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

## Acknowledgements

We are indebted to Prof. Satoshi Yoshida, Waseda University, for making a number of helpful suggestions. We thank Mayumi Nakazawa, Masashi Abe, Takahiro Kudo, Fuune Inoue, Yuki Imaizumi, and Toshihiko Deguchi for their assistance with a series of experiments. Gratitude is also extended to the Gene Research Center for Radioisotopes at Hirosaki University for the use of the facilities. This work was supported by a Grant-in-Aid for Scientific Research (C) from the Ministry of Education, Culture, Sports, Science, and Technology of Japan [grant number 15K06945] (to N.N.), a grant from the Naito Foundation of Japan (to N.N.), and a grant from the Institute for Fermentation, Osaka (IFO) (to N.N.).

## Author contributions

S.H. M.T., and N.N. designed the study. S.H., R.K., Y.S., M.W., and M.T. performed experiments using *S. cerevisiae*. R.K. performed experiments using the in vitro translation system. S.H., D.K., and H.H. performed radioisotope experiments. S.H. and N.N. wrote the manuscript; M.T., M.W. D.K., and H.H. revised it.

## Competing interests

The authors declare no competing interests.
