## [Peer Review File · Communications Biology]

Reviewers' comments:

Reviewer #1 (Remarks to the Author):

The article by Hoshino et al describes the role of two stalled ribosome rescue factors (srRF) in antibiotic resistance in yeast. To demonstrate their findings, the authors used antibiotics both bacteriocidal and bacteriostatic, that are known to stall ribosomes. The authors concluded that one of the srRF (Ylr281) plays a crucial role in successfully rescuing stalled antibiotic bound ribosomes. Although the article is well written with proper background there are some technical issues that are listed below:

1) Results: Page 7: "at 30c, the *yol114cΔ* and *ylr281cΔ* grew slower than the WT..." However, in fig 1b: the growth curves showed that at 30c all mutants (except the double mutant" showed similar growth? Please explain

2) Fig 1b: *ylr281cΔ* labelled as "blown" should be "brown"? Also why in Fig 1b in presence of Tc the growth of double mutant is not shown? Is the double mutant in presence of Tc did not grow at all? There are some colonies visible in the YPG plate at 30C at concentration 1 in Fig 1c for the double mutant suggesting some growth? Please explain.

3) Page 7: while *yol114cΔylr281cΔ* mutant grew slightly slower than the wild type (Fig 1b) Is this growth rate difference statistically significant? Were the growth curves performed in biological replicates?

4) Results: Page 9: The deletion of YOL114C had a slight effect on growth, whereas that of YLR281C had a significant effect."... However, in Fig 1c: the growth defects in both the mutants looks similar for tetracycline, doxycycline, erythromycin and chloramphenicol. Only difference in the growth between the two mutants is visible in azithromycin. So how is it that the growth defects for *ylr281cΔ* is "significant" while *yol114cΔ* is "slight" when compared to the WT? A better experiment to show the difference would be to do Minimum Inhibitory concentration (MIC) to these drugs in the mutants. Otherwise it is quite difficult to justify the statement. Please explain. To the reviewer, both the mutants have slight growth defects in presence of the drugs when compared to the wild type.

5) Results Page 10: "...the susceptibility phenotype ... was not suppressed by plasmid borne YOL114C and YLR281C mutants having VAQ residues instead of GGQ motifs...." However, the spot assays in presence of Tc and Em looks similar between GGQ and VAQ motif constructs. Further, VAQ motif constructs for both the genes grew better in presence of Tc and Em when you compare them with the growth of the mutants alone in Fig 3a. For example, *yol114cΔ* mutant grows till 1/2 conc in presence of Tc while the construct *yol114cΔ+VAQ* showed colonies till 1/100 dilution. This shows that VAQ is still been able to restore growth than the mutant alone. Please explain. Again a better experiment to demonstrate significant differences between the VAQ and GGQ motifs would be to do MIC assays with the drugs.

6) Fig 3a) From the spot assays, the Em susceptibility between the mutants and the wild type is not visibly different. The author should consider using Azithromycin instead of erythromycin for the assay. Please explain

7) Why did the authors used two different concentrations of tetracycline for spot assays (100ug/ml) and for mitochondrial protein expression/FACS (500ug/ml). Please explain? For consistency same concentration of drug needs to be used.

8) Fig 2: *ylr281cΔ* labelled as "blown" should be "brown"?

9) How much concentration of Tc used to analyze the gene expression by qPCR? Please mention it in the methods and also in the legend of Fig 4.

10) Methods: Page 19: Please mention the final concentration of Mitotracker REd and green used in the experiment?

11) Fig 2: The authors found difference in mitochondrial mass between the mutants and the wild type in YPG media. Did the authors used YPD as a control to see if there is a difference in mass between the mutant in the YPD media to begin with? please explain

12) Methods: Page 20: In the SDS-PAGE experiment to determine the expression of the mitochondrial proteins, how did the authors confirm that they loaded equal amount of protein in the gel. In other words what did the authors used as a loading control to determine that they loaded same amount of total protein in the gel?

13) Methods: Page 20: qPCR: How was the RNA purity checked? How much RNA was used to quantify the expression? Was DNase step performed to get rid of genomic DNA contaminants? How was the data analysed? Please explain

Reviewer #2 (Remarks to the Author):

In this manuscript, Hoshino et al. examine *Saccharomyces cerevisiae* Yol114c, the ortholog of the human stalled ribosome rescue factor (srRF) ICT1, and Ylr281c, the ortholog of human C12orf65, which appears to be important for mitochondria homeostasis in humans and *S. pombe* and may functionally overlap with ICT1, although direct evidence for it acting as a srRF is lacking. The authors demonstrate that loss of either YOL114C or YLR281C reduced yeast cell viability under respiratory growth conditions relative to wild type and the addition of heat stress (37°C) leads to synthetic lethality when both genes are absent, indicating they may have parallel or redundant functions in mitochondria homeostasis. By flow cytometry, loss of YOL114C and YLR281C, alone or together, increases mitochondrial mass without altering the membrane potential, which may indicate the increase in mass is compensatory. The authors go on to show that loss of YLR281C more significantly sensitizes cells to bacteriostatic antibiotics that cause mitoribosome stalling. Expressing excess YOL114C in *ylr281Δ* cells does not fully suppress this sensitivity (while the converse does) and mutation of the putative GGQ motif of YLR281C that would be required for peptidyl-tRNA hydrolysis (PTH) activity, also leads to a slight increase in antibiotic sensitivity. When treated with bacteriostatic antibiotics, cells lacking Ylr281c also grow slower in liquid media, do not show the same increase in mitochondrial mass, and, importantly, show a decrease in mitochondrial translation products without a change in transcription. Finally, the authors show that Yol114c and Ylr281c trigger PTH in vitro. This study is generally well-done with high quality data and most of the conclusions being justified by the data shown. The authors establish for the first time that a C12orf65 ortholog, Ylr281c, has PTH activity for a non-stop template. They also establish that Ylr281c is uniquely required for viability and mitochondrial translation during bacteriostatic antibiotic treatment, suggesting that it is critically required for dealing with this type of stress. These findings may have implications for antibiotic resistance in fungi, making this study also of interest to those outside of the field of translation. However, there are some flaws that would need to be addressed prior to publication. The values presented for the PTH activity, particularly for a no-go template, are largely lacking primary data and statistics to justify these numbers. The authors also fail to link the PTH activity/rescue activity that they demonstrate in vitro to the antibiotic sensitivity of cells and effects on mitochondrial translation. These data seem crucial to their conclusion that Ylr281c functions as a rescue factor for antibiotic-dependent stalled ribosomes.

Major Comments:

1. As stated above, the authors do not convincingly link the PTH activity of Ylr281c to the antibiotic sensitivity and effects on mitochondrial translation. This should be addressed in several ways:
 - a. The current data presented in Figure 3b regarding the VAQ mutant does not include the deletion strains expressing only vector alone (pRS316), which is needed to compare the degree of antibiotic sensitivity and confirm that there really is no enhancement of viability by the VAQ mutant. Also, there does not appear to be a difference in the antibiotic sensitivity of the Ylr281c and Yol114c VAQ mutants in Figure 3b, whereas the two deletion mutants in Figure 1c under the same conditions have a growth difference. How do the authors explain this discrepancy?

- b. The authors should demonstrate that their VAQ mutant phenocopies the *ylr281Δ* in their mitochondrial in vitro translation analysis (Figure 4a) and does not stimulate hydrolysis in their PTH assay (Figure 5a).
2. The lack of full suppression of the antibiotic growth defect of *ylr281cΔ* by *Yol114c* expressed from a centromeric plasmid under control of its native promoter could be related to the relative levels of the two proteins under endogenous conditions and not because of differing stalling specificities. The authors should assess whether over-expressing *Yol114c* from a stronger promoter or from a higher copy number plasmid fully suppresses the growth defect.
3. Without drug, it appears as though all mitochondrial protein translation products are increased in the deletions relative to wild type, and not just *Var1*. Is this reproducible? The mitochondrial protein synthesis data in Figure 4a should be quantified for each protein/band and normalized to the levels in wild type mitochondria for at least three replicates.
4. Where did the 50% relative activity values for the No-go template in Figure 5b come from? Where is the primary data and relative activity graph for this experiment (as shown in Figure 5a for non-stop template)? Also, the 50% activity numbers in Figure 5b table should be presented as mean with STD for replicates. Further, it is not appropriate to include data for *YaeJ* from a published study in Figure 5a. Either remove this data from the figure or repeat it as a true control in the context of this experiment.

Minor Comments:

1. The use of the systematic ORF names (*Ylr281c* and *Yol114c*) when standard names exist for an ORF (*Rso55* and *Pth4*) is not common. Is there a reason the authors have chosen to do so, and if not, it would be preferable to use the standard names for these genes.
2. The order of the figures is not the order that they are discussed in the text, meaning the reader has to go ahead and back in the figures as they read the text. It would be helpful to the reader to have the figures ordered linearly in the way they are discussed in the text.
3. Page 4, line 20: Typo of *ICY1* rather than *ICT1*.
4. Page 6, line 4: *Ylr281* is missing the "c" at the end.
5. Page 7, line 13: The authors say that double mutant, "grew either only slightly more slowly than the mutants or at the same rate..." Which is it, or was there variability from experiment to experiment, and if so, can the authors explain this?
6. Figure 1b and 2: the red line for *ylr281Δ* is labeled as "blown" rather than brown (or red, which is more accurate).
7. Figure 2 legend: Please include the temperature for the FCM.
8. Figure 3 legend indicates that these experiments were done on YPG plates with drug. Why was this not done on synthetic medium (SG) to maintain selection for the indicated plasmids?

Reviewer #3 (Remarks to the Author):

This paper characterizes two mitochondrial proteins' role in response to antibiotics and defects in mitoribosome translation. *Pth4* and *Rso55* have been previously identified as mitochondrial proteins. In the mutants, the membrane potential did not change but the size of the mitochondria increased. The authors reason this is to compensate for less than fully functional ribosomes. In general, *rso55* mutants were more sensitive to bacterial antibiotics than *pth4* with the double mutants being the most sensitive. However, the levels of ROS did not change between the mutants, but the level of mitochondrial translation did decrease, without a significant change in mRNA levels of mitochondrial transcripts. In an in vitro assay, the addition of purified truncated *Rso5* and *Pth4* proteins restarted stalled mitochondrial ribosomes.

1. In general, proteins and genes should be referred to by their standard names and only referred to by their systematic (ORF) names when they have not yet been named. Table 1 has the standard names, Ptr4 and Rso55, and should be used instead of systematic names.

2. The introduction is disjointed going from one species to another and back and forth between the two proteins. There was a noticeable lack of information about *S. cerevisiae* homologs, Ptr4 and Rso55, the two proteins that are the focus of this paper. Ptr4 had been purified associating with mitoribosomes (Kehrein et al. 2015). Deletion of pth4 inhibits mitochondrial translation in vivo (Woellhaf et al. 2016). Rso55 has also been predicted to be a ribosomal protein (Perocchi et al. 2006) and purified with mitochondrial ribosomes (Morgenstern et al. 2017).

3. In Fig 1a, serial dilutions are presented to demonstrate differences in mutant growth, but this assay is best for drastic differences and not for differences less than 2-fold. In Figure 1b, you showed that the double mutant grew slower; however, there are no error bars so no quantitative conclusions can be drawn. I recommend liquid growth assays with at minimum three biological replicates. *S. cerevisiae* grown YPG and measured plate readers can reduce the work but yeast grow very slowly and may not show the growth differences.

4. Fig 1c needs a loading control for the plating such as YPD grown at 30C and a control for YPG plates otherwise not much can be said about relative growth with the double mutant growing so slowly in different antibiotics. The controls reported in supplemental fig are from a different experiment as they are not the same dilution series can't be used to directly compare growth among the different antibiotic containing plates. In Fig 1c, it is hard to tell without the YPG plate to see how much growth is affected in the mutants. I can roughly estimate from figure 1a, that the double mutant doesn't grow anyway so adding an antibiotic can't make a dead strain grow any less.

5. On page 7-8, text describing the flow cytometry should be moved to when Fig 2 is discussed which 38 should be after Fig 1 is discussed. Jumping back and forth between figures disrupts the flow of the paper to the reader.

6. In Fig 3a, qRT-PCR and/ or western blots would confirm the plasmid borne, RSO55 or PTH4 are expressed. Do expression levels of RSO55 or PTH4 change when yeast are treated with antibiotics? From (Ho et al. 2018), Rso55 mean protein levels 731 (coefficient of variance 132) and Pth4 mean protein levels are 619 (coefficient of variance 132). Did the authors ensure that both constructs are expressed at similar levels?

7. Plating control with the same dilutions need to be shown for Fig 3b. From here, it looks like Gp1 (VAQ) and Gp2 (VAQ) might have less of an effect. I would prefer a liquid growth assay as more quantitative assay but at the bare minimum, the control plate needs to be shown.

8. In Fig 4a, the coomassie stain of the gel looks like rso55 mutant in -Tc has more protein while in +Tc it looks like it has slightly less but that would not account for the much-reduced translation in +Tc but could in -Tc. In reading the methods, there is no internal normalization for total protein loaded in the analysis of mitochondrial translation products. So, the authors need to temper their conclusions without a loading control.

9. In Fig 4a, rso55 mutant grows much slower in +Tc in so wouldn't any mutant that slows growth in these conditions also slow mitochondria translation?

10. Serial plate assays only measure the lack of growth which could be from growth arrest or from loss

of cellular viability. After exposure to the two different types of antibiotics, can the mutants recover their normal growth? Does acute exposure reduce viability or are cells arrested and then can return to normal growth once the antibiotics are removed? There was no measurable difference in ROS so cells may arrest their growth without reducing viability.

11. Would petite cells with no mitochondrial DNA be more resistant to these antibiotics than grande yeast?

12. Based on the markers, I assume that MTY3015 is the BY4741 strain. However, MTY3016 markers do not correspond to BY4742. Then how was the diploid selected to cross rso55 and pth4 mutants into the same strain?

13. Rso55 "rescued dependent stalled mitoribosomes, which appear to represent a "no-go" type of ribosomes stalled on intact mRNA." While this is likely based on the genetic analysis, the in vitro assay was not done with antibiotics to conclusively support this conclusion.

Minor

14. In all figures with graphs, was "blown" supposed to be "brown" such as Fig 1a, 2, and 4?

15. Page 5, third paragraph, second sentence is vague. "The binding of antibiotics to ribosomes may create a type that differs from a non-stop type because antibiotic-bound ribosomes are stalled somewhere on intact fulllength mRNAs." What is the "type" referring to? At the end of the page, the following sentence, "Many of the antibiotics that bind to bacterial ribosomes bind to mitoribosomes, not cytoplasmic ribosomes, to impair cell viability and function through mitochondrial dysfunctions, and, in the case of clinical treatments, sometimes cause severe side effects." Are the cytoplasmic ribosomes the eukaryotic 80S ribosomes?

16. From the following sentence, it is unclear what the results are without deciphering them from the figure. "At 37°C, the *yol114Δ* and *y1r281Δ* mutants grew slower than the wild type, whereas the *yol114Δy1r281Δ* mutant appeared to be lethal (Fig. 1a)." I recommend the following edits, "At 37°C, the *yol114Δ* and *y1r281Δ* single mutants grew slower than the wild type in YPG, whereas the *yol114Δy1r281Δ* double mutant appeared to be lethal but are viable in YPD (Fig. 1a)."

17. The following sentence from page 9 is unclear to me, "Thus, difficulties are associated with assessing the proper functions of the two srRFs using cells grown in the presence of bactericidal antibiotics." Are you referring to difficulties in the previously mentioned study, or difficulties in this study of Pth4 and Rso55?

18. Gut microbiome can secrete antibiotics. Here is a review.

<https://www.ncbi.nlm.nih.gov/pubmed/29018846>

19. For the fluorescent detection of mitochondrial characteristics, what was the final OD of the culture? Were the cells kept in log-phase? If cells were in stationary phase would that impact the mitochondrial characteristics that were measured?

20. Why was it necessary to truncate Rso55 and Pth4 at the N-terminal? Did you ensure that these would be functional proteins?

References

Ho, B., A. Baryshnikova, and G. W. Brown, 2018 Unification of Protein Abundance Datasets Yields a Quantitative *Saccharomyces cerevisiae* Proteome. *Cell Syst.* 6: 192-205.e3.

Kehrein, K., R. Schilling, B. V. Möller-Hergt, C. A. Wurm, S. Jakobs et al., 2015 Organization of

Mitochondrial Gene Expression in Two Distinct Ribosome-Containing Assemblies. *Cell Rep.* 843–853.

Morgenstern, M., S. B. Stiller, P. Lübbert, C. D. Peikert, S. Dannenmaier et al., 2017 Definition of a High Confidence Mitochondrial Proteome at Quantitative Scale. *Cell Rep.* 19: 2836–2852.

Perocchi, F., L. J. Jensen, J. Gagneur, U. Ahting, C. Von Mering et al., 2006 Assessing systems properties of yeast mitochondria through an interaction map of the organelle. *PLoS Genet.* 2: 1612–1624.

Woellhaf, M. W., F. Sommer, M. Schroda, and J. M. Herrmann, 2016 Proteomic profiling of the mitochondrial ribosome identifies Atp25 as a composite mitochondrial precursor protein. *Mol. Biol. Cell* 27: 3031–3039.

Reviewer #1 (Remarks to the Author):

The article by Hoshino et al describes the role of two stalled ribosome rescue factors (srRF) in antibiotic resistance in yeast. To demonstrate their findings, the authors used antibiotics both bacteriocidal and bacteriostatic, that are known to stall ribosomes. The authors concluded that one of the srRF (Ylr281) plays a crucial role in successfully rescuing stalled antibiotic bound ribosomes. Although the article is well written with proper background there are some technical issues that are listed below:

Thank you for your insightful comments and suggestions.

First, please refer to Fig. 2 in the revised version, in which results have been obtained by additional experiments. As suggested by all reviewers, previous results using the plasmids expressing the mutants having VAQ residues were obscure or misleading. We think that the main reason for the unclear results is that a single-gene deletion mutant keeps the other gene expressed or functional. Thus, for clarification, we have conducted the corresponding experiments again using the double-gene deletion strain instead of the single-gene deletion strains. The following results have become obvious (Fig. 2). (1) In the presence of tetracycline, chloramphenicol or azithromycin, growth defect of the double-gene deletion strain can be fully suppressed by plasmid born *ylr281c* (*pth3/rso55*), whereas that can be only partially suppressed by plasmid born *yol114c* (*pth4*). (2) That cannot be suppressed by the VAQ mutants. These results strongly support the findings that we have obtained thus far.

Note that we use “*PTH4*” and “*PTH3*” instead of *YOL114C* and *YLR281C*, respectively, in the revised manuscript.

1) Results: Page 7: "at 30°C, the *yol114cΔ* and *ylr281cΔ* grew slower than the WT..." However, in fig 1b: the growth curves showed that at 30c all mutants (except the double mutant" showed similar growth? Please explain

Figures 1a and 1b in the previous version were misleading. We have performed some experiments again and replaced the figures with each revised version (Fig. 1a (plates) and Fig. 3a (liquid)). In the spot assay (Fig. 1a), dilution rates were changed and the spot number was increased for more clarification. In growth curves, we have repeated measurements of OD600 at three time points six times, followed by Student's test.

The results on the YPG plates in the absence of an antibiotic showed that at 30°C, the *yol114cΔ* and *ylr281cΔ* single mutants grew similarly to the wild type. As a result, no essential differences have been observed in growth of the single-gene deletion mutants between solid and liquid media.

Thus, we have modified the sentence as below.

Page 7; line 11: “At 30°C, the *yol114cΔ* and *ylr281cΔ* single mutants grew similarly to the wild type”

2) Fig 1b: *ylr281cΔ* labelled as "blown" should be "brown"? Also why in Fig 1b in presence of Tc the growth of double mutant is not shown? Is the double mutant in presence of Tc did not grow at all? There are some colonies visible in the YPG plate at 30C at concentration 1 in Fig 1c for the double mutant suggesting some growth? Please explain.

This labelling is our mistake, so we have corrected it.

As suggested, we have measured growth of the double-gene deletion mutant in liquid media on the presence of Tc, and added the results to Fig. 3a and Supplementary Fig 5a. The results showed severely impaired growth rate, but not lethality, of the double mutant.

We have added a sentence about the growth of the double-gene deletion mutant (*pth4Δpth3Δ*) in liquid medium, as below.

Page 10; line 16: “whereas that of the *pth4Δpth3Δ* mutant was severely impaired (Fig. 3a, Supplementary Fig. 5a).”

3) Page 7: while *yol114cΔylr281cΔ* mutant grew slightly slower than the wild type (Fig 1b) Is this growth rate difference statistically significant? Were the growth curves performed in biological replicates?

As mentioned above, we have repeated measurement of OD600 of each strain at three time points six times, and performed statistical analysis by Student's t test (Fig. 3a).

According to the results, we have modified the sentence as below.

“a small decrease in growth of the *pth4Δpth3Δ* (*yol114cΔylr281cΔ*) mutant compared to that of the wild type was only observed in the exponential growth phase (Fig. 3a and Supplementary Fig. 5a).”

4) Results: Page 9: The deletion of YOL114C had a slight effect on growth, whereas that of YLR281C had a significant effect."... However, in Fig 1c: the growth defects in both the mutants looks similar for tetracycline, doxycycline, erythromycin and chloramphenicol. Only difference in the growth between the two mutants is visible in azithromycin. So how is it that the growth defects for *ylr281cΔ* is "significant" while

yol114 Δ is "slight" when compared to the WT? A better experiment to show the difference would be to do Minimum Inhibitory concentration (MIC) to these drugs in the mutants. Otherwise it is quite difficult to justify the statement. Please explain. To the reviewer, both the mutants have slight growth defects in presence of the drugs when compared to the wild type.

I am worried that it might be difficult to discriminate differences in growth in Fig. 1 when the reviewer prints the figure of the PDF file on a sheet of paper and examines yeast spots because image contrast or resolution sometimes depends on the type of printer.

As suggested, "significant" or "slight" were not appropriate and the words have been omitted. We examined adequate sublethal concentration of each antibiotic instead of MIC, at which differences of growth among the mutants and the wild-type were clarified (Supplementary Fig. 3). From the results of repeated experiments of spot assay, we concluded that growth of yol114 Δ is slower than that of ylr281 Δ in the presence of the antibiotics on each concentration indicated.

We have modified the sentence as below.

Page 8; line 8: The order of decrease in growth rate appears to be *pth4* Δ *pth3* Δ > *pth3* Δ > *pth4* Δ (*yol114* Δ *ylr281* Δ > *ylr281* Δ > *yol114* Δ).

5) Results Page 10: ..."the susceptibility phenotype ... was not suppressed by plasmid borne YOL114C and YLR281C mutants having VAQ residues instead of GGQ motifs..." However, the spot assays in presence of Tc and Em looks similar between GGQ and VAQ motif constructs. Further, VAQ motif constructs for both the genes grew better in presence of Tc and Em when you compare them with the growth of the mutants alone in Fig 3a. For example, yol114 Δ mutant grows till 1/2 conc in presence of Tc while the construct yol114 Δ +VAQ showed colonies till 1/100 dilution. This shows that VAQ is still been able to restore growth than the mutant alone. Please explain. Again a better experiment to demonstrate significant differences between the VAQ and GGQ motifs would be to do MIC assays with the drugs.

The response to this comment was described in the beginning of this letter. We have modified the sentences according to the results of additional experiments, which range from page 8, line 28 to page 10, line 1.

6) Fig 3a) From the spot assays, the Em susceptibility between the mutants and the wild type is not visibly different. The author should consider using Azithromycin instead of

erythromycin for the assay. Please explain

In new experiments (Fig. 2), we have used azithromycin instead of erythromycin, and in addition, chloramphenicol according to your comment.

7) Why did the authors used two different concentrations of tetracycline for spot assays (100 µg/ml) and for mitochondrial protein expression/FACS (500 µg/ml). Please explain? For consistency same concentration of drug needs to be used.

It is because sublethal concentrations of Tc appear to differ between liquid and solid media. For example, in the presence of 500 µg/mL of Tc, growth of the wild type can be observed for 72 h in liquid medium (Supplementary Fig. 5, which has been created with results of additional experiments); however that can hardly be observed for 72 h in solid medium (Supplementary Fig. 3). In the presence of 100 µg /mL of Tc in liquid medium, no substantial differences of growth were observed between the *ylr281cΔ* mutant and the wild-type (Supplementary Fig. 5). Thus, we selected 500 µg/mL of Tc in liquid media. Such differences in the effects of gene deletion on growth rates between liquid and solid media have sometimes been observed in yeast (1-3).

We have added a sentence about the difference in concentrations of Tc as below

Page 10, line 20: In the presence of Tc, the concentration in liquid medium was increased to 500 µg/mL so that differences of growth between the mutants and the wild type could be clarified (Supplementary Fig. 5b).

1. Kuzmenko, A. et al. (2016) Aim-less translation: loss of *Saccharomyces cerevisiae* mitochondrial translation initiation factor mIF3/Aim23 leads to unbalanced protein synthesis. *Sci Rep*, 6, 18749.
2. Bauerschmitt, H. et al. (2008) The membrane-bound GTPase Guf1 promotes mitochondrial protein synthesis under suboptimal conditions. *J Biol Chem*, 283, 17139-17146.
3. Van Hoof, C. et al. (2000) The *Saccharomyces cerevisiae* homologue YPA1 of the mammalian phosphotyrosyl phosphatase activator of protein phosphatase 2A controls progression through the G1 phase of the yeast cell cycle. *J Mol Biol*, 302, 103-120.

8) Fig 2: *ylr281cΔ* labelled as "blown" should be "brown"?

This is our mistake, so we have corrected it.

9) How much concentration of Tc used to analyze the gene expression by qPCR? Please mention it in the methods and also in the legend of Fig 4.

The concentration of Tc was 500 $\mu\text{g/ml}$ because the samples were grown in liquid media. We have added the information into the legend of Fig. 4 (page 24, line 18).

10) Methods: Page 19: Please mention the final concentration of Mitotracker Red and green used in the experiment?

A stock solution of each dye is 1 mM according to the manufacturer's procedure, and so the final concentrations of the two dyes are 1 μM . We have added this information to the Methods section.

Page 19, line19: "1 μL of 1 mM MitoTracker Red or Green was added"

11) Fig 2: The authors found difference in mitochondrial mass between the mutants and the wild type in YPG media. Did the authors used YPD as a control to see if there is a difference in mass between the mutant in the YPD media to begin with? please explain.

As long as glucose is supplied, a budding yeast prefers fermentation to aerobic respiration even in the presence of oxygen. This kind of fermentative capacity allows growth of yeast in YPD medium not to require mitochondrial oxidative phosphorylation or mtDNA (namely mitochondrial translation). So, we think that the FCM experiments using YPD media do not work well as the control.

12) Methods: Page 20: In the SDS-PAGE experiment to determine the expression of the mitochondrial proteins, how did the authors confirm that they loaded equal amount of protein in the gel. In other words what did the authors used as a loading control to determine that they loaded same amount of total protein in the gel?

Before RI labeling, protein concentrations from isolated mitochondria (mg/mL) were determined with the Bradford assay (Biorad) according to previous papers (for example, Garcia-Cazarin, 2011; Divakaruni, 2013). By reference to the values, we loaded equal amounts of mitochondria from the three strains to the gel.

We have added a sentence below about how to determine protein concentrations to the Methods section.

Page 21, line 4: After each isolated mitochondrial pellet was mixed with 1 mL of SH buffer, protein concentration (mg/mL) was determined with the Bradford assay (Bio-Rad).

Garcia-Cazarin, M.L. et al. Mitochondrial isolation from skeletal muscle. *J. Vis. Exp.* (2011).

Divakaruni, A.S. et al. Thiazolidinediones are acute, specific inhibitors of the mitochondrial pyruvate carrier. *Proc. Natl. Acad. Sci. U. S. A.* 110, 5422-5427 (2013).

13) Methods: Page 20: qPCR: How was the RNA purity checked? How much RNA was used to quantify the expression? Was DNase step performed to get rid of genomic DNA contaminants? How was the data analysed? Please explain

The RNA quality has been checked by formaldehyde agarose gel electrophoresis, and the results have been shown in Supplementary Fig. 9.

1 ng of total RNA was used for qPCR.

RNA was extracted from cells of each strain using the PureLink RNA Mini Kit (Invitrogen), which includes a treatment step with DNase.

The qPCR results were analyzed using a LightCycler 480 system software based on $\Delta\Delta C_t$ method.

We have added this information to the Methods section (qPCR: Page 20, lines 21-27).

Reviewer #2 (Remarks to the Author):

In this manuscript, Hoshino et al. examine *Saccharomyces cerevisiae* Yol114c, the ortholog of the human stalled ribosome rescue factor (srRF) ICT1, and Ylr281c, the ortholog of human C12orf65, which appears to be important for mitochondria homeostasis in humans and *S. pombe* and may functionally overlap with ICT1, although direct evidence for it acting as a srRF is lacking. The authors demonstrate that loss of either YOL114C or YLR281C reduced yeast cell viability under respiratory growth conditions relative to wild type and the addition of heat stress (37°C) leads to synthetic lethality when both genes are absent, indicating they may have parallel or redundant functions in mitochondria homeostasis. By flow cytometry, loss of YOL114C and YLR281C, alone or together, increases mitochondrial mass without altering the membrane potential, which may indicate the increase in mass is compensatory. The authors go on to show that loss of YLR281C more significantly sensitizes cells to bacteriostatic antibiotics that cause mitoribosome stalling. Expressing excess YOL114C in *ylr281*^Δ cells does not fully suppress this sensitivity (while the converse does) and mutation of the putative GGQ motif of YLR281C that would be required for peptidyl-tRNA hydrolysis (PTH) activity, also leads to a slight increase in antibiotic sensitivity. When treated with bacteriostatic antibiotics, cells lacking Ylr281c also grow slower in liquid media, do not show the same increase in mitochondrial mass, and, importantly, show a decrease in mitochondrial translation products without a change in transcription. Finally, the authors show that Yol114c and Ylr281c trigger PTH in vitro. This study is generally well-done with high quality data and most of the conclusions being justified by the data shown. The authors establish for the first time that a C12orf65 ortholog, Ylr281c, has PTH activity for a non-stop template. They also establish that Ylr281c is uniquely required for viability and mitochondrial translation during bacteriostatic antibiotic treatment, suggesting that it is critically required for dealing with this type of stress. These findings may have implications for antibiotic resistance in fungi, making this study also of interest to those outside of the field of translation. However, there are some flaws that would need to be addressed prior to publication. The values presented for the PTH activity, particularly for a no-go template, are largely lacking primary data and statistics to justify these numbers. The authors also fail to link the PTH activity/rescue activity that they demonstrate in vitro to the antibiotic sensitivity of cells and effects on mitochondrial translation. These data seem crucial to their conclusion that Ylr281c functions as a rescue factor for antibiotic-dependent stalled ribosomes.

Thank you for your insightful comments and suggestions.

First, please refer to Fig. 2 in the revised version, in which results have been obtained by additional experiments. As suggested by all reviewers, previous results using the plasmids expressing the mutants having VAQ residues were obscure or misleading. We

think that the main reason for the unclear results is that a single-gene deletion mutant keeps the other gene expressed or functional. Thus, for clarification, we have conducted the corresponding experiments again using the double-gene deletion strain instead of the single-gene deletion strains. The following results have become obvious (Fig. 2). (1) In the presence of tetracycline, chloramphenicol or azithromycin, growth defect of the double-gene deletion strain can be fully suppressed by plasmid born *ylr281c* (*pth3/rso55*), whereas that can be only partially suppressed by plasmid born *yol114c* (*pth4*). (2) That cannot be suppressed by the VAQ mutants. These results strongly support the findings that we have obtained thus far.

Note that we use “*PTH4*” and “*PTH3*” instead of *YOL114C* and *YLR281C*, respectively, in the revised manuscript.

Major Comments:

1. As stated above, the authors do not convincingly link the PTH activity of Ylr281c to the antibiotic sensitivity and effects on mitochondrial translation. This should be addressed in several ways:

a. The current data presented in Figure 3b regarding the VAQ mutant does not include the deletion strains expressing only vector alone (pRS316), which is needed to compare the degree of antibiotic sensitivity and confirm that there really is no enhancement of viability by the VAQ mutant. Also, there does not appear to be a difference in the antibiotic sensitivity of the Ylr281c and Yol114c VAQ mutants in Figure 3b, whereas the two deletion mutants in Figure 1c under the same conditions have a growth difference. How do the authors explain this discrepancy?

b. The authors should demonstrate that their VAQ mutant phenocopies the *ylr281c* in their mitochondrial *in vitro* translation analysis (Figure 4a) and does not stimulate hydrolysis in their PTH assay (Figure 5a).

a) The response to this comment was described in the beginning of this letter. Additional experiments include those using the strain expressing only the vector alone (pRS316) according to your comment.

We have modified the sentences according to the results of additional experiments, which range from page 8, line 28 to page 10, line 1.

b) Additional experiments have shown that neither VAQ mutant of Ylr281c nor that of Yol114c has any ability to suppress growth defect of the double-gene deletion mutant. Besides, as shown in Supplementary Fig 8c, both VAQ mutant of Ylr281c and Yol114c exhibited even lower PTH activities compared to those of the corresponding wild type proteins although in the *E. coli in vitro* translation. Thus, we conclude that the VAQ

mutants have no detectable PTH activity in mitochondria.

2. The lack of full suppression of the antibiotic growth defect of *ylr281c* by *Yol114c* expressed from a centromeric plasmid under control of its native promoter could be related to the relative levels of the two proteins under endogenous conditions and not because of differing stalling specificities. The authors should assess whether over-expressing *Yol114c* from a stronger promoter or from a higher copy number plasmid fully suppresses the growth defect.

As suggested, we have done additional experiments, in which the native promoter was replaced with a strong constitutive *TEF1* promoter, in the *Yol114c* expression vector (Supplementary Fig 4b). However, it caused a decrease rather than an increase in the suppression efficiency using the *TEF1* promoter than the native promoter. This result showed that the lack of full suppression of the antibiotic-driven growth defect of *ylr281c* cannot be compensated for simply by an increase of protein expression of *Yol114c*.

Furthermore, we have performed additional qPCR experiments, which showed that *Tc* did not affect gene expression of *YOL114C* (*PTH4*) or *YLR281C* (*PTH3*) from the native promoter in the wild type (Supplementary Fig 5c).

We have added a sentence about the experiments of the promoter replacement as below.
Page 9, line 18: Additionally, to check if the partial suppression was improved by more increased gene expression of *PTH4*, we replaced the native promoter with a strong constitutive *TEF1* promoter in pRS316 Gp 114C (Supplementary Fig. 4b). However, it caused a decrease rather than an increase in the suppression efficiency using the *TEF1* promoter than the native promoter.

Page 12, line 28: Additionally, we confirmed if *Tc* affected gene expression of *PTH4* or *PTH3* in the wild type. No significant changes were detected in expression of either genes (Supplementary Fig. 5c).

3. Without drug, it appears as though all mitochondrial protein translation products are increased in the deletions relative to wild type, and not just *Var1*. Is this reproducible? The mitochondrial protein synthesis data in Figure 4a should be quantified for each protein/band and normalized to the levels in wild type mitochondria for at least three replicates.

We had already performed the experiments three times. As suggested, we have quantified the mitochondrial protein synthesis data band by band using ImageQuant

LAS 4010, and calculated SD for error bars (Supplementary Fig. 7b). As a result, no substantial increase of mitochondrial protein products except Var1 were observed in the absence of Tc.

4. Where did the 50% relative activity values for the No-go template in Figure 5b come from? Where is the primary data and relative activity graph for this experiment (as shown in Figure 5a for non-stop template)? Also, the 50% activity numbers in Figure 5b table should be presented as mean with STD for replicates. Further, it is not appropriate to include data for YaeJ from a published study in Figure 5a. Either remove this data from the figure or repeat it as a true control in the context of this experiment.

As suggested, data in Figure 5b in a previous version was not appropriate. We have replaced it with a new graph format, in which concentrations of the two proteins required for approximately 60% PTH activity were indicated (Fig. 5b). In this format, error bars (SD) can be shown. The gel image in Figure 5b in a previous version has been added to Supplementary Fig. 8e.

We have modified the sentence as below.

Page 14, line 16: In contrast, approximately 60% PTH activity of Pth3 for non-stop and no-go mRNAs required similar concentrations (3.0 μ M and 4.5 μ M, respectively).

As suggested, data for YaeJ was omitted in Figure 5a.

Minor Comments:

1. The use of the systematic ORF names (Ylr281c and Yol114c) when standard names exist for an ORF (Rso55 and Pth4) is not common. Is there a reason the authors have chosen to do so, and if not, it would be preferable to use the standard names for these genes.

Ylr281c and Yol114c proteins are homologous in terms of sequence and structure. We think that if the names “Rso55” and “Pth4” are used, it would be more difficult for readers to understand this homology. Additionally, *RSO55* was named after a mitochondrial protein Related to Spastic paraplegia type 55 with Optic atrophy and neuropathy (*SPG55*), which is one of three alias names of *C12orf65*. In human, the gene name, *C12orf65*, is most often used in papers, whereas *SPG55* is rarely used. Because there are no apparent connections among spastic paraplegia, the number “55” and yeast *YLR281C*, we would like to use *PTH3* instead of *RSO55*, which is named after *pth3*, the corresponding gene name in *S. pombe*.

We have added a sentence below to the Introduction section.

Page 5, line 18: In this study, we refer to *RSO55* as *PTH3* to avoid confusion between the species.

2. The order of the figures is not the order that they are discussed in the text, meaning the reader has to go ahead and back in the figures as they read the text. It would be helpful to the reader to have the figures ordered linearly in the way they are discussed in the text.

As suggested, we have changed the order of figures (Fig. 2 and Fig. 3), and transferred Fig 1b (OD600 measurements, in a previous version) to Fig. 3 (FCM measurements). Accordingly, we have modified the text.

3. Page 4, line 20: Typo of ICY1 rather than ICT1.

We have corrected it.

4. Page 6, line 4: Ylr281 is missing the “c” at the end.

We have corrected it.

5. Page 7, line 13: The authors say that double mutant, “grew either only slightly more slowly than the mutants or at the same rate...” Which is it, or was there variability from experiment to experiment, and if so, can the authors explain this?

Data at 30°C in Figure 1a was not clear and the sentence was ambiguous. For clarification, we have changed dilution rates and growth periods in the spot assay (Fig. 1a). Besides, we have performed additional experiments using liquid media, in which OD600 of each strain had been measured at three time points six times, and done statistical analysis by Student's t test (Fig. 3a).

We have modified the sentence as below.

Page 7; line 11: At 30°C, the *pth4Δ* and *pth3Δ* single mutants grew similarly to the wild type, while the *pth4Δpth3Δ* double mutant grew slightly slower than the wild type (Fig. 1a).

6. Figure 1b and 2: the red line for *ylr281c* is labeled as “blown” rather than brown (or red, which is more accurate).

This is our mistake, so we have corrected it.

7. Figure 2 legend: Please include the temperature for the FCM.

FCM measurements were done at room temperature (about 25°C) in the measurement room that was air-conditioned 24/7. We have added this information into the legend.

Page 24, line 4: “Mitochondrial mass (left) and membrane potential (right) in the wild type (WT), *pth4Δ*, *pth3Δ*, and *pth4Δpth3Δ* grown in liquid YPG media for 48 h were measured at room temperature (about 25°C) by FCM using....”

8. Figure 3 legend indicates that these experiments were done on YPG plates with drug.

We have modified the legend of Figure 2 (corresponding to Figure 3 in a previous version).

Page 23, line 21: “Each strain was diluted and spotted onto SG-ura plates containing no antibiotic, 100 µg/mL tetracycline,”

9. Why was this not done on synthetic medium (SG) to maintain selection for the indicated plasmids?

As suggested, SG media are better than YPG media for maintaining selection for the indicated plasmids. Thus, we have used SG-ura media instead of YPG media in all experiments using strains transformed with each plasmid (Fig. 2).

We have also modified the sentences as below.

Page 9, line 6: This plasmid also carries *URA3* as an auxotrophic marker gene.

Page 9, line 7: “the spot assay on synthetic medium plates lacking uracil and containing 3% glycerol (SG-ura) showed”

Reviewer #3 (Remarks to the Author):

This paper characterizes two mitochondrial proteins' role in response to antibiotics and defects in mitoribosome translation. Pth4 and Rso55 have been previously identified as mitochondrial proteins. In the mutants, the membrane potential did not change but the size of the mitochondria increased. The authors reason this is to compensate for less than fully functional ribosomes. In general, *rso55* mutants were more sensitive to bacterial antibiotics than *pth4* with the double mutants being the most sensitive. However, the levels of ROS did not change between the mutants, but the level of mitochondrial translation did decrease, without a significant change in mRNA levels of mitochondrial transcripts. In an in vitro assay, the addition of purified truncated Rso5 and Pth4 proteins restarted stalled mitochondrial ribosomes.

Thank you for your insightful comments and suggestions.

First, please refer to Fig. 2 in the revised version, in which results have been obtained by additional experiments. As suggested by all reviewers, previous results using the plasmids expressing the mutants having VAQ residues were obscure or misleading. We think that the main reason for the unclear results is that a single-gene deletion mutant keeps the other gene expressed or functional. Thus, for clarification, we have conducted the corresponding experiments again using the double-gene deletion strain instead of the single-gene deletion strains. The following results have become obvious (Fig. 2). (1) In the presence of tetracycline, chloramphenicol or azithromycin, growth defect of the double-gene deletion strain can be fully suppressed by plasmid born *ylr281c* (*pth3/rso55*), whereas that can be only partially suppressed by plasmid born *yol114c* (*pth4*). (2) That cannot be suppressed by the VAQ mutants. These results strongly support the findings that we have obtained thus far.

Note that we use “*PTH4*” and “*PTH3*” instead of *YOL114C* and *YLR281C*, respectively, in the revised manuscript.

1. In general, proteins and genes should be referred to by their standard names and only referred to by their systematic (ORF) names when they have not yet been named. Table 1 has the standard names, *Ptr4* and *Rso55*, and should be used instead of systematic names.

Ylr281c and *Yol114c* proteins are homologous in terms of sequence and structure. We think that if the names “*Rso55*” and “*Pth4*” are used, it would be more difficult for readers to understand this homology. Additionally, *RSO55* was named after a mitochondrial protein Related to Spastic paraplegia type 55 with Optic atrophy and Neuroopathy (*SPG55*), which is one of three alias names of *C12orf65*. In human, the gene

name, *C12orf65*, is most often used in papers, whereas *SPG55* is rarely used. Because there are no apparent connections among spastic paraplegia, the number “55” and yeast *YLR281C*, we would like to use *PTH3* instead of *RSO55*, which is named after *pth3*, the corresponding gene name in *S. pombe*.

We have added a sentence below to the Introduction section.

Page 5, line 18: In this study, we refer to *RSO55* as *PTH3* to avoid confusion between the species.

2. The introduction is disjointed going from one species to another and back and forth between the two proteins. There was a noticeable lack of information about *S. cerevisiae* homologs, Ptr4 and Rso55, the two proteins that are the focus of this paper. Ptr4 had been purified associating with mitoribosomes (Kehrein et al. 2015). Deletion of *pth4* inhibits mitochondrial translation in vivo (Woellhaf et al. 2016). Rso55 has also been predicted to be a ribosomal protein (Perocchi et al. 2006) and purified with mitochondrial ribosomes (Morgenstern et al. 2017).

As suggested, we have added sentences below to the Introduction section.

Page 5, line 16: As for fungi, homologous genes of *ICT1* and *C12orf65* have been found in the fission yeast *Schizosaccharomyces pombe* (*pth4* and *pth3*) and the budding yeast *Saccharomyces cerevisiae* (*PTH4* and *RSO55*, respectively) (Table 1).

Page 5, line23: Although comprehensive proteomic analysis of mitochondrial proteins from *S. cerevisiae* confirmed that Pth4 is present in a ribosome-bound and in a free form in mitochondria^{24, 25}, while Pth3 is localized in mitochondria²⁶, little is known about the physiological and biochemical functions of the two proteins in mitochondria.

Note that we have cited a paper by Woellhaf et al. (2016) regarding the localization of Pth4, but refrained from citing the paper regarding the function of Pth4 (deletion of *PTH4* inhibits mitochondrial translation in vivo) for the following reasons. i) In Figure 2b in the paper, the lane corresponding to Δ pth4 strain indeed shows no detectable bands of eight proteins coded by mtDNA, but shows the presence of a band of Cox2 coded by mtDNA as an internal control in western blots. ii) The result was obtained using cells grown on fermentable carbon sources, whereas we focus the function of Pth4 in cells grown in non-fermentable carbon sources. iii) We cannot have information about how to make deletion mutants and the genetic background of the mutants because supplementary data is not available on the web. According to previous papers by the same authors, their mutants seem to be derived from the W303 wild-type strain, which is different from ours (BY4741).

3. In Fig 1a, serial dilutions are presented to demonstrate differences in mutant growth, but this assay is best for drastic differences and not for differences less than 2-fold. In Figure 1b, you showed that the double mutant grew slower; however, there are no error bars so no quantitative conclusions can be drawn. I recommend liquid growth assays with at minimum three biological replicates. *S. cerevisiae* grown YPG and measured plate readers can reduce the work but yeast grow very slowly and may not show the growth differences.

As suggested, we have done liquid growth assay and made bar graphs with error bars (SD, n= 6) (Fig. 3a) followed by a statistical analysis using Student's t-test. Additionally, for clarification, we have repeated the spotting assay again, in which dilution series were changed such that they are identical in all the plates (Fig. 1a).

We have modified the sentences as below.

Page 10, line 8: In the absence of Tc, growth rates were essentially the same between the two single-gene deletion mutants and the wild type, while a small decrease in growth of the pth4 Δ pth3 Δ mutant compared to that of the wild type was only observed in the exponential growth phase (Fig. 3a and Supplementary Fig. 5a).

4. Fig 1c needs a loading control for the plating such as YPD grown at 30°C and a control for YPG plates otherwise not much can be said about relative growth with the double mutant growing so slowly in different antibiotics. The controls reported in supplemental fig are from a different experiment as they are not the same dilution series can't be used to directly compare growth among the different antibiotic containing plates. In Fig 1c, it is hard to tell without the YPG plate to see how much growth is affected in the mutants. I can roughly estimate from figure 1a, that the double mutant doesn't grow anyway so adding an antibiotic can't make a dead strain grow any less.

As suggested, loading controls (YPD at 30°C and 37°C in the absence of an antibiotic) have been added to Fig 1a. The same dilution series have been used in the spot assays between Figs 1a and b.

As indicated, in the presence of antibiotics, the dilution series of spot assay on YPG plates in Supplementary Fig. 2 differ from those on YPD plates in Fig 1b. However, in Supplementary Fig. 2, no substantial differences of growth in YPD between the mutants and the wild type were observed. Thus, none of the antibiotics at indicated concentrations (Fig. 1b (in a previous version, Fig. 1c)) had any effects on cell viability of the mutants or the wild type. Note that OD600 of the starting samples was always 1.0 in all spot assays in YPD and YPG media.

In the absence of an antibiotic, at 30°C, the double mutant grew slightly slower than the wild type, whereas, at 37°C, it could hardly grow. All experiments in the presence of antibiotics were performed at 30°C.

5. On page 7-8, text describing the flow cytometry should be moved to when Fig 2 is discussed which 38 should be after Fig 1 is discussed. Jumping back and forth between figures disrupts the flow of the paper to the reader.

As suggested, we have changed the order of figures (Fig. 2 and Fig. 3), and transferred Fig 1b (OD600 measurements, in a previous version) to Fig. 3 (FCM measurements). Accordingly, we have modified the text.

6. In Fig 3a, qRT-PCR and/ or western blots would confirm the plasmid borne, RSO55 or PTH4 are expressed. Do expression levels of RSO55 or PTH4 change when yeast are treated with antibiotics? From (Ho et al. 2018), Rso55 mean protein levels 731 (coefficient of variance 132) and Pth4 mean protein levels are 619 (coefficient of variance 132). Did the authors ensure that both constructs are expressed at similar levels?

We have done additional qPCR experiments to examine whether the presence of Tc in YPG medium affected expression of YOL114C (PTH4) and YLR281C (PTH3/RSO55). No significant changes were observed in expression of YOL114C or YLR281C at a transcriptional level between the presence and absence of Tc (Supplementary Fig 5c). Thus, it appears that none of the native promoters of the two genes are Tc-inducible. Because the promoters in the plasmids are also native ones, presumably, the protein expression levels of plasmid-borne genes are similar irrespective of the presence or absence of Tc based on data of Ho et al.

We have added a sentence below about additional qPCR experiments.

Page 13, line 2: Additionally, we confirmed if Tc affects gene expression of YOL114C or YLR281C. No significant changes were detected in gene expression of either genes (Supplementary Fig. 5c).

7. Plating control with the same dilutions need to be shown for Fig 3b. From here, it looks like Gp1 (VAQ) and Gp2 (VAQ) might have less of an effect. I would prefer a liquid growth assay as more quantitative assay but at the bare minimum, the control plate needs to be shown.

The response to this comment was described in the beginning of this letter. In additional

experiments, appropriate control plates have been shown (Fig. 2).

8. In Fig 4a, the coomassie stain of the gel looks like *rso55* mutant in -Tc has more protein while in +Tc it looks like it has slightly less but that would not account for the much-reduced translation in +Tc but could in -Tc. In reading the methods, there is no internal normalization for total protein loaded in the analysis of mitochondrial translation products. So, the authors need to temper their conclusions without a loading control.

Before RI labeling, total protein concentrations from isolated mitochondria (mg/mL) were determined with the Bradford assay (Biorad) according to previous papers (for example, Garcia-Cazarin, 2011; Divakaruni, 2013). By reference to the values, we loaded equal amounts of total mitochondrial proteins from the three strains to the gel. As suggested, because this experiments have no internal control protein, we have weakened the conclusion.

Garcia-Cazarin, M.L. et al. Mitochondrial isolation from skeletal muscle. *J. Vis. Exp.* (2011).

Divakaruni, A.S. et al. Thiazolidinediones are acute, specific inhibitors of the mitochondrial pyruvate carrier. *Proc. Natl. Acad. Sci. U. S. A.* 110, 5422-5427 (2013).

We have added a sentence below about how to determine protein concentrations to the Methods section.

Page 21, line 4: After each isolated mitochondrial pellet was mixed with 1 mL of SH buffer, protein concentration (mg/mL) was determined with the Bradford assay (Bio-Rad).

In the Abstract section:

Page 2, line 15: “showed” -> “suggested”

In the Results and Discussion sections:

We have modified a sentence as below

Page 12, line 10: “in the presence of Tc, all eight protein products was significantly decreased” -> “appeared to be decreased”

We have added a sentence as below.

Page 12, line 16: However, the possibility cannot be excluded that the reduction rates in protein products differ from those in vivo.

We have modified a sentence as below

Page 15, line 24: “indicated that in the presence of Tc, the deletion of *YLR281C*, but not *YOL114C*, markedly decreased” -> “suggested that in the presence of Tc, the deletion of *YLR281C*, but not *YOL114C*, decreased”

9. In Fig 3a (4?), *rso55* mutant grows much slower in +Tc in so wouldn't any mutant that slows growth in these conditions also slow mitochondria translation?

There are few experiments using YPG media in the presence of Tc, and, to our knowledge, no such mutants have been reported.

10. Serial plate assays only measure the lack of growth which could be from growth arrest or from loss of cellular viability. After exposure to the two different types of antibiotics, can the mutants recover their normal growth? Does acute exposure reduce viability or are cells arrested and then can return to normal growth once the antibiotics are removed? There was no measurable difference in ROS so cells may arrest their growth without reducing viability.

To clarify if the cause of such growth defects is due to growth arrest or loss of cell viability, we have performed additional experiments according to the comment (Supplementary Fig. 6b). The results showed that growth defect in the presence of sublethal concentrations of each antibiotics is due to growth arrest.

We have added a sentence below to the Results section.

Page 11, line 11: We also examined whether the growth impairment of the mutants in the presence of Tc is due to growth arrest or loss of cellular viability. The mutants were grown in YPG liquid media containing Tc, paromomycin or azithromycin for 1 h, and were washed with PBS buffer to remove each antibiotic. The samples were spotted on YPG plates in the absence of the antibiotics, showing that the mutants grew as well as the wild type (Supplementary Fig. 6b). Combined with the results of the FCM analysis, these results suggest that the antibiotics at each sublethal concentration arrest cell growth of the mutants rather than reduce cell viability.

11. Would petite cells with no mitochondrial DNA be more resistant to these antibiotics than grande yeast?

When grown in YPD media, such petite cells might be more resistant to mitoribosome-target antibiotics. However, petite cells cannot intrinsically grow on

nonfermentable carbon sources (YPG media). We have never observed petite cells of the wild type or the three mutants on YPD or YPG plates in the presence of the antibiotics at sublethal concentrations.

12. Based on the markers, I assume that MTY3015 is the BY4741 strain. However, MTY3016 markers do not correspond to BY4742. Then how was the diploid selected to cross *rso55* and *pth4* mutants into the same strain?

Instead of selection by nutrient markers, we physically isolated the diploid cell by micromanipulation as follows. In brief, cells of two strains having opposite mating types (e.g., MTY3462 and MTY3459) were mixed on a YPD plate and left for 4-5 h at 30°C. After streaking the mixture on a new plate, individual zygotes were identified by visual inspection under a microscope (CX31, Olympus, Japan) and isolated one by one using a micromanipulator (Narishige, Japan) equipped with the microscope. Zygotes are T- or dumbbell-shaped, which can be easily distinguished from haploid cells.

We have added a sentence below to the Methods section.

Page 18, line 23: Tetrad spores were separated under a microscope (CX31, Olympus, Japan) using a micromanipulator (Narishige, Japan).

13. *Rso55* “rescued dependent stalled mitoribosomes, which appear to represent a “no-go” type of ribosomes stalled on intact mRNA.” While this is likely based on the genetic analysis, the *in vitro* assay was not done with antibiotics to conclusively support this conclusion.

It is difficult to do experiments such that an antibiotic stalls ribosomes on a specific position of intact mRNA. Ribosome stalling on a specific position needs to be detected in the gel. We had attempted to do experiments in which Tc was added to ribosomes that had already been stalled on a stop codon of intact mRNA (no-go ribosomes), as described in the text. Expectedly, PTH activity of *Yol281c* (*Rso55*) were not changed.

We have added sentences explaining why we used no-go ribosomes in the absence of antibiotics.

Page 14, line 2: “We attempted to test whether *Ylr281c* and *Yol114c* rescue no-go ribosomes caused by an antibiotic using the *in vitro* translation system. However, an antibiotic works at miscellaneous sites on intact mRNA, so that it would produce heterogenous no-go ribosomes, leading to the lack of a discrete band on the gel. Instead, ...”

Minor

14. In all figures with graphs, was “blown” supposed to be “brown” such as Fig 1a, 2, and 4?

This is our mistake, so we have corrected it.

15. Page 5, third paragraph, second sentence is vague. “The binding of antibiotics to ribosomes may create a type that differs from a non-stop type because antibiotic-bound ribosomes are stalled somewhere on intact full length mRNAs.” What is the “type” referring to? At the end of the page, the following sentence, “Many of the antibiotics that bind to bacterial ribosomes bind to mitoribosomes, not cytoplasmic ribosomes, to impair cell viability and function through mitochondrial dysfunctions, and, in the case of clinical treatments, sometimes cause severe side effects.” Are the cytoplasmic ribosomes the eukaryotic 80S ribosomes?

Page 5, line 31: For clarification, “a type” has been changed to “a stalled-ribosome type”

As indicated, the cytoplasmic ribosomes are the eukaryotic 80S ribosomes. We have added “eukaryotic” to the sentence.

Page 6, line 6: “cytoplasmic ribosomes” -> “eukaryotic cytoplasmic ribosomes”

16. From the following sentence, it is unclear what the results are without deciphering them from the figure. “At 37°C, the *yol114cΔ* and *ylr281cΔ* mutants grew slower than the wild type, whereas the *yol114cΔylr281cΔ* mutant appeared to be lethal (Fig. 1a).” I recommend the following edits, “At 37°C, the *yol114cΔ* and *ylr281cΔ* single mutants grew slower than the wild type in YPG, whereas the *yol114cΔylr281cΔ* double mutant appeared to be lethal but are viable in YPD (Fig. 1a).”

As suggested, we have modified the sentence according to the comment.

Page 7, line 15: At 37°C, as observed at 30°C, the *pth4Δ* and *pth3Δ* single mutants grew as well as the wild type in YPG media, whereas the *pth4Δpth3Δ* double mutant appeared to be lethal in YPG but viable in glucose-containing media (YPD) (Fig. 1a).

17. The following sentence from page 9 is unclear to me, “Thus, difficulties are associated with assessing the proper functions of the two srRFs using cells grown in the presence of bactericidal antibiotics.” Are you referring to difficulties in the previously

mentioned study, or difficulties in this study of Pth4 and Rso55?

As suggested, this sentence was obscure; we refer to difficulties in this study of Pth4 and Pth3/Rso55. We have modified the sentence as below.

Page 8, line 22: Thus, difficulties would be associated with assessing the proper functions of the two proteins as a rescue factor of stalled ribosomes in mitochondria if using cells grown in the presence of bactericidal antibiotics.

18. Gut microbiome can secrete antibiotics. Here is a review.
<https://www.ncbi.nlm.nih.gov/pubmed/29018846>

We thank you for the useful information. At present, no such mitoribosome-binding antibiotics have been reported that are secreted from microbiomes.

We have modified the sentence as below.

Page 17, line 6: “Since human cells are apparently not attacked by such antibiotics as used in this study, except in the case of therapeutic use,”

19. For the fluorescent detection of mitochondrial characteristics, what was the final OD of the culture? Were the cells kept in log-phase? If cells were in stationary phase would that impact the mitochondrial characteristics that were measured?

Cell samples used in FCM measurements were grown in YPG media and harvested after 48 hours. Growth of cells in YPG always requires respiration and mitochondrial gene expression, and does not undergo a diauxic shift. In the presence of Tc, cells were in the late exponential phase, and, in the absence of Tc, cells were in the beginning of the stationary phase of growth (Supplementary Fig. 5a). In the absence of Tc, we measured FCM using cells harvested after 24 h, which were around the exponential phase, and we confirmed an increase in mitochondrial mass of the mutants, as described in the text, although its mass showed a smaller increase after 24 h than for 48 h. Although the possibility cannot be excluded that there is some influence on mitochondrial characteristics of cells in YPG, depending on difference of growth phases, we think that there is no need to change the conclusion in this case.

Page 10, line 21: We have added “in cells grown in liquid YPG media for 48 h” into the Results section and the legend of Fig. 3.

20. Why was it necessary to truncate Rso55 and Pth4 at the N-terminal? Did you ensure that these would be functional proteins?

The N-terminal truncated regions were expected to be a mitochondrial-targeting signal (See the Method section). Thus, remove of the region is thought to help prevent loss or decrease of the protein function. At least in the in vitro translation, purified truncated proteins showed enough PTH activities to detect in the gel.

References

Ho, B., A. Baryshnikova, and G. W. Brown, 2018 Unification of Protein Abundance Datasets Yields a Quantitative *Saccharomyces cerevisiae* Proteome. *Cell Syst.* 6: 192-205.e3.

Kehrein, K., R. Schilling, B. V. Möller-Hergt, C. A. Wurm, S. Jakobs et al., 2015 Organization of Mitochondrial Gene Expression in Two Distinct Ribosome-Containing Assemblies. *Cell Rep.* 843–853.

Morgenstern, M., S. B. Stiller, P. Lübbert, C. D. Peikert, S. Dannenmaier et al., 2017 Definition of a High Confidence Mitochondrial Proteome at Quantitative Scale. *Cell Rep.* 19: 2836–2852.

Perocchi, F., L. J. Jensen, J. Gagneur, U. Ahting, C. Von Mering et al., 2006 Assessing systems properties of yeast mitochondria through an interaction map of the organelle. *PLoS Genet.* 2: 1612–1624.

Woellhaf, M. W., F. Sommer, M. Schroda, and J. M. Herrmann, 2016 Proteomic profiling of the mitochondrial ribosome identifies Atp25 as a composite mitochondrial precursor protein. *Mol. Biol. Cell* 27: 3031–3039.

Reviewers' comments:

Reviewer #1 (Remarks to the Author):

The authors satisfactorily answered to all my queries and added new good quality data.

Reviewer #2 (Remarks to the Author):

The authors have done a good job of addressing the concerns of myself and the other reviewers and the manuscript will be suitable for publication once two minor corrections are made.

1. There are still errors in the use of "Blown" rather than "Brown" that needs to be corrected (Figures 3a and 3b and Supplementary Figures 5a, 5b, and 6a).

2. Figure 4 and Supplementary Figure 7b should have student's t-tests done and P-values displayed to confirm insignificance (Figure 4) or significance (Supplementary Figure 7b).

Reviewer #3 (Remarks to the Author):

Reviewer #3

The authors have added requested controls and rewritten sections that warranted attention.

1. Before you publish Rso55 as Pth3, I suggest contacting Saccharomyces Genome Database to update the gene page. Using non-standard gene names adds to the confusion in the field. In the past and SGD has worked to improve consistency across publications¹. Typically the first to publish has precedence in adoption of standard gene names but there are cases when another gene name can become standard.

2-5. Changes were made.

6. Supplementary Figure 5c is discussed after supplemental figure 7 and 6. It should be moved reflect the order of figures discussed in the paper or move the results in the text to reflect the order of the figures.

7. Figure 2. pRS316 Gp1 114C contains PTH4 under the endogenous promoter. I suggest maintaining uniformity in the nomenclature and refer to this plasmid as pPTH4 rather than Gp1 in the figure unless there is something particular about the gene that I am missing. pRS316 Gp2 281c contains PTH3 under the endogenous promoter. I suggest maintaining uniformity in the nomenclature and refer to this plasmid as pPTH3 rather than Gp2 unless there is something particular about the gene that I am missing. Gp1(VAG) and Gp2(VAG) mutants can be labeled pPTH4(VAG) and pPTH3(VAG). Figure S4 pRS316 Tp1 114C contains PTH4 under the TEF1 promoter. I suggest maintaining uniformity in the nomenclature and refer to this plasmid as pTEFpr-PTH4 rather than Tp1 in the figure unless there is something particular about the gene that I am missing. pRS316 Tp2 281c contains PTH3 under the TEF1 promoter. I suggest maintaining uniformity in the nomenclature and refer to this plasmid as pTEF1pr-PTH3 rather than Tp2 unless there is something particular about the gene that I am missing.

8-10. Changes were made.

11. Yes, of course.

12-13.

14. Figures 3, S5, and S6 still have blown instead of brown.

15-20. Changes were made.

21. In the methods, give the amino acid numbers that correspond to the GGQ motif in Pth3 and Pth4 because it is not easily to determine that from Figure S1.

22. Cells expressing only PTH4 are more sensitive to tetracycline and really more sensitive to chloramphenicol. While over-expression of PTH3 increases sensitivity to antibiotics rather improve resistance. That could be from negative feedback regulation from increased levels of PTH3 causing downregulation of PTH4 translation or changes stability of Pth4 protein itself. The authors did rule out changes in PTH4 mRNA levels. The other possibility is that Pth3 does bind the mitoribosomes but doesn't function as well as Pth4 and blocks another unknown release factor such as Mrf1 from binding and relieving the blocked ribosomes. While I have no suggestions to further modify the paper, these are possibilities that could be addressed in the discussion.

1. Appendix A1: Yeast Nomenclature Systematic Open Reading Frame (ORF) and Other Genetic Designations. in *Alternative pre-mRNA Splicing 603–607* (Wiley-VCH Verlag GmbH & Co. KGaA, 2012). doi:10.1002/9783527636778.app1

Reviewer #2 (Remarks to the Author):

The authors have done a good job of addressing the concerns of myself and the other reviewers and the manuscript will be suitable for publication once two minor corrections are made.

Thank you very much for your comments.

1. There are still errors in the use of "Blown" rather than "Brown" that needs to be corrected (Figures 3a and 3b and Supplementary Figures 5a, 5b, and 6a).

I am truly sorry that you have pointed out the mistake twice. I have corrected it.

2. Figure 4 and Supplementary Figure 7b should have student's t-tests done and P-values displayed to confirm insignificance (Figure 4) or significance (Supplementary Figure 7b).

As suggested, we have statistical analysis by Student's t test. All p-values are shown in each table included in Fig. 4 and Supplementary Fig. 7b.

Reviewer #3 (Remarks to the Author):

The authors have added requested controls and rewritten sections that warranted attention.

Thank you very much for your comments.

1. Before you publish Rso55 as Pth3, I suggest contacting Saccharomyces Genome Database to update the gene page. Using non-standard gene names adds to the confusion in the field. In the past and SGD has worked to improve consistency across publications¹. Typically the first to publish has precedence in adoption of standard gene names but there are cases when another gene name can become standard.

As suggested, I sent an email about this alias gene name to the SGD curators about one week ago, but I have not got any reply yet.

I understand that the assigned gene name should be used, and appreciate the first researchers that named the gene. However, I am concerned about future trouble in consistency across the species rather than publications. We have also used Rso55 in the Abstract and Introduction sections.

We consult the SGD curators about the use of Pth3 and will follow their advice as much as possible.

2-5. Changes were made.

6. Supplementary Figure 5c is discussed after supplemental figure 7 and 6. It should be moved reflect the order of figures discussed in the paper or move the results in the text to reflect the order of the figures.

As suggested, Supplemental figure 5c has been moved to Supplemental figure 7.

7. Figure 2. pRS316 Gp1 114C contains PTH4 under the endogenous promoter. I suggest maintaining uniformity in the nomenclature and refer to this plasmid as pPTH4 rather than Gp1 in the figure unless there is something particular about the gene that I am missing. pRS316 Gp2 281c contains PTH3 under the endogenous promoter. I suggest maintaining uniformity in the nomenclature and refer to this plasmid as pPTH3 rather than Gp2 unless there is something particular about the gene that I am missing. Gp1(VAG) and Gp2(VAG) mutants can be labeled pPTH4(VAG) and pPTH3(VAG). Figure S4 pRS316 Tp1 114C contains PTH4 under the TEF1 promoter. I suggest maintaining uniformity in the nomenclature and refer to this plasmid as pTEFpr-PTH4 rather than Tp1 in the figure unless there is something particular about the gene that I am missing. pRS316 Tp2 281c contains PTH3 under the TEF1 promoter. I suggest maintaining uniformity in the nomenclature and refer to this plasmid as pTEF1pr-PTH3 rather than Tp2 unless there is something particular about the gene that I am missing.

Thank you for your comment. We have changed all corresponding nomenclatures in the text and figures (Fig. 2 and Supplementary Fig. 4) according to your suggestion.

8-10. Changes were made.

11. Yes, of course.

12-13.

14. Figures 3, S5, and S6 still have blown instead of brown.

I am truly sorry that you have pointed out the mistake twice. I have corrected it.

15-20. Changes were made.

21. In the methods, give the amino acid numbers that correspond to the GGQ motif in Pth3 and Pth4 because it is not easily to determine that from Figure S1.

As suggested, we have added a sentence to the Methods section.

Page 19, line 3: The pPTH4(VAQ) and pPTH3(VAQ) plasmids in which the GGQ motif residues (Gly80-Gly81-Gln82 and Gly49-Gly50-Gln51) (Supplementary Fig. 1) were changed to Val-Ala-Gln in pPTH4 and pPTH3, respectively, were prepared by site directed mutagenesis.

22. Cells expressing only PTH4 are more sensitive to tetracycline and really more sensitive to chloramphenicol. While over-expression of PTH3 increases sensitivity to antibiotics rather improve resistance. That could be from negative feedback regulation from increased levels of PTH3 causing downregulation of PTH4 translation or changes stability of Pth4 protein itself. The authors did rule out changes in PTH4 mRNA levels. The other possibility is that Pth3 does bind the mitoribosomes but doesn't function as well as Pth4 and blocks another unknown release factor such as Mrf1 from binding and relieving the blocked ribosomes. While I have no suggestions to further modify the paper, these are possibilities that could be addressed in the discussion.

Figure 2 shows that cells expressing only PTH4 are more sensitive to tetracycline and really more sensitive to chloramphenicol. Supplementary Figure 4b shows that

over-expression of PTH3 increases sensitivity to antibiotics rather improve resistance. However, because these results were obtained using the double-gene deletion mutant, it would be unlikely that increased levels of PTH3 cause downregulation of PTH4 transcription or translation.

We think that there are other possibilities as you suggested. We have added sentences below to the Results section.

Page 9, line 21: It was also found that overexpression of Pth3 considerably impaired growth even in the absence of Tc (Supplementary Fig. 4b). It is possible that an excessive amount of either of srRFs causes its unexpected binding to normal translating mitoribosomes, preventing normal elongation or termination.

1. Appendix A1: Yeast Nomenclature Systematic Open Reading Frame (ORF) and Other Genetic Designations. in *Alternative pre-mRNA Splicing 603–607* (Wiley-VCH Verlag GmbH & Co. KGaA, 2012). doi:10.1002/9783527636778.app1